# Improved cookstoves to reduce household air pollution exposure in sub-Saharan Africa: A scoping review of intervention studies

**Eunice Phillip**[1]*, **Jessica Langevin**[1], **Megan Davis**[1], **Nitya Kumar**[2], **Aisling Walsh**[1], **Vincent Jumbe**[3], **Mike Clifford**[4], **Ronan Conroy**[1], **Debbi Stanistreet**[1]

**1** Department of Public Health and Epidemiology, School of Population Health, Royal College of Surgeons in Ireland, University of Medicine and Health Sciences, Dublin, Ireland, **2** Department of Medicine, Royal College of Surgeons in Ireland University of Medicine, and Health Sciences -Bahrain, Manama, Bahrain, **3** Department of Health Systems and Policy, Kamuzu University of Health Sciences, Mahatma Gandhi, Blantyre, Malawi, **4** Faculty of Engineering, University of Nottingham, Nottingham, United Kingdom

* eunicephillip@rcsi.ie

**Data Availability Statement:** All data and processes files are available from the Open Science

## Abstract

Household air pollution (HAP), primarily from biomass fuels used for cooking, is associated with adverse health outcomes and premature mortality. It affects almost half of the world's population, especially in low-income and low-resourced communities. However, many of the 'improved' biomass cookstoves (ICS) aimed at reducing HAP lack empirical evidence of pollutant reduction and reliability in the field. A scoping review guided by the Joanna Briggs Institute framework was systematically conducted to explore and analyse the characteristics of cookstoves to assess the ICS available to meet the socio-economic and health needs of households in sub-Sahara Africa (sSA). The review searched Scopus, PubMed, Web of Science, EMBASE, Global Health Database on OVID, BASE, and conducted a grey literature search from 2014 to 2022 for all field-based ICS studies. In addition, user perspectives were explored for cookstoves analysed as available, affordable, and effective in reducing harmful biomass emissions. The search returned 1984 records. Thirty-three references containing 23 ICS brands were included. The cookstoves were analysed into seven categories: (1) efficiency in HAP reduction, (2) availability, (3) affordability, (4) sustainability, (5) safety, (6) health outcomes, and (7) user experience. Most (86.9%) of the improved cookstoves showed a reduction in harmful emission levels compared to the traditional three-stone fire. However, the levels were higher than the WHO-recommended safe levels. Only nine were priced below 40 USD. Users placed emphasis on cookstoves' suitability for cooking, fuel and time savings, safety, and price. Equality in cooking-related gender roles and psychosocial benefits were also reported. The review demonstrated limited field testing, a lack of evidence of ICS emissions in real-life settings in sSA, heterogeneity in emission measurements, and incomplete descriptions of ICS and kitchen features. Gender differences in exposure and psychosocial benefits were also reported. The review recommends improved cookstove promotion alongside additional measures to reduce HAP at a cost affordable to low-resource households. Future research should focus on detailed reporting of study parameters to facilitate effective comparison of ICS performance in different social

Framework repository (https://doi.org/10.17605/OSF.IO/PKZGH).

**Funding:** This review is part of The Smokeless Village Project, funded by the Irish Research Council, project number: COALESCE/2020/13 awarded to author DS. The funder had no role in the study design, collection, analysis, interpretation of data, and in the decision to submit the article for publication.

**Competing interests:** The authors have declared that no competing interests exist.

settings with different local foods and fuel types. Finally, a more community-based approach is needed to assess and ensure user voices are represented in HAP intervention studies, including designing the cookstoves.

## Introduction

The incomplete combustion of biomass fuel and kerosene in traditional cookstoves (TCS) and with three-stone fires (TSF) used in low-income communities emits pollutants such as fine particulate matter ($PM_{2.5}$) and toxic gases, including carbon monoxide (CO), contributing to household air pollution (HAP) and ambient air pollution [1, 2]. These pollutants are linked to environmental damage through deforestation [3], climate depleting compounds such as black carbon (BC) [4]. They are also directly associated with poor health outcomes, including pregnancy-related complications [5, 6], cardiovascular and respiratory illnesses [7–9].

The World Health Organization (WHO) described HAP as one of the most significant environmental health risks, accounting for 7.7% of global mortality in 2016 [10]. This includes 25% of chronic obstructive pulmonary disease, 12% of strokes, 17% of lung cancers, 45% of pneumonia-related deaths in children under five years old (CU5) [4], and a higher risk of burn injuries [4, 11]. While this is of global concern, the burden is highest in low- and middle-income countries (LMICs) [8] among the poorest rural communities that rely on biomass fuel due to lack of access (availability and affordability) to cleaner energy sources and technologies [4, 10]. Women and children in these communities bear the highest burden, accounting for 60% of all HAP-related deaths [11, 12], with seven times the levels of $PM_{2.5}$ and CO in women and adolescent girls compared to men and boys [13]. In addition, time lost to women and girls due to fuel collection further exacerbates the gender inequality gap and cycle of poverty [14].

While these health and social issues could be addressed with cleaner energy sources (e.g., liquid petroleum gas (LPG) and electricity) and advanced cookstove technologies (e.g., solar-powered and biogas cookstoves), these technologies remain unavailable, inaccessible, and unaffordable. In addition, the barriers to uptake and adoption of interim HAP-reducing practices (e.g., ventilation and behavioural practices such as reducing time spent in proximity to the open fire and improved cookstoves) also remain a challenge. Similarly, the adoption of the widely promoted improved cookstoves has been hindered by its inability to meet users' cooking needs compared with the traditional three-stone fire (TSF) [15, 16], as well as other factors affecting financial and market development [15, 17, 18], regulation, standards, and quality of ICS [18, 19].

Our initial review of the literature demonstrated a wealth of studies measuring improved cookstove emissions [20–22] and health outcomes [23–25], including systematic reviews [7, 9, 15, 16, 18, 26–28]. In addition, the effectiveness of these promoted cookstoves has mainly been tested in laboratory conditions [17, 19]. However, there remains limited research on ICS effectiveness in reducing HAP in real-life settings. This makes it difficult to benchmark their emission performance with the 2014 WHO indoor air quality (WHO-IAQ) guidelines [29] in real-life field situations as these values neither reflect how ICS perform in actual settings [30] nor their performance across different cultures and cooking practices.

Further, where available, studies tend to focus primarily on emission levels, and they often lack information on the cookstoves' affordability, availability, sustainability, and ability to meet user cooking needs. Therefore, this review seeks to address this gap and build on the recommendation of Stanistreet et al. [30] to bring together all relevant information on improved

cookstoves and review field-tested ICS evidence in sub-Saharan Africa (sSA), given that this region will account for 90% of the world's population without electricity by 2030. This review thus aims to identify which improved cookstoves would be the most suitable to promote among poor communities in sub-Saharan Africa.

## Materials and methods

This scoping review (ScR) examines evidence of improved cookstoves' effectiveness, efficiency, and benefit to the poorest populations in sSA. Scoping reviews provide overviews of a topic, synthesise and map existing evidence, and identify gaps in a subject area [31, 32]. The protocol for this study was developed using the Joanna Briggs Institute's (JBI) six-step framework for ScR. Originally developed by Arksey and O'Malley [32], the framework was modified by Levac and colleagues [33] and enhanced by Peters et al. [34]. The protocol was registered prospectively on 'The Open Science Framework', available at https://osf.io/pkzgh/. Our reporting structure was guided by the Preferred Reporting Items for Systematic Reviews and Meta-Analysis extension for Scoping Reviews (PRISMA-ScR) [35] (S1 Table). All the review stages were iterative to ensure full exploration and understanding of the literature's content.

### The six-step framework

**1. Identifying and developing the review questions.** The research questions, inclusion and exclusion criteria were guided by JBI's Participants/Population, Concepts, Context (PCC) search framework (Table 1) [34]. In consultation with social scientist experts, the review questions were developed to explore the state of current ICS evidence (Concept) that could be promoted among the poorest populations in sSA (Population) to reduce HAP and improve health outcomes (Context) (Table 1).

**2. Identifying relevant studies and eligibility criteria.** Six electronic databases were initially searched with the collaboration of an experienced librarian in June 2020. EP and DS screened the first 100 records for titles and abstracts to refine the terms used in the subsequent search of Scopus, PubMed, Web of Science, Embase, the Global Health Database on OVID, and BASE. EP and JL manually searched relevant websites (S1B Fig) and individual records within systematic reviews, and carried out backward snowballing of studies eligible for full-text screening. The search was updated in July 2021 and September 2022 (S1A Fig).

ICS were considered as an intervention if described as having the potential to reduce pollutants and improve health outcomes compared to the TSF (pot placed on three stone over an open fire) [36] or traditional cookstove (locally made from mud or metal and slightly more fuel-efficient than the three-stone fire) [37]. Studies eligibility criteria are as follows;

**Table 1. Review questions and domains developed with JBI's PCC and experts' consultation.**

| Review questions | Domains |
|---|---|
| 1. What are the ICS characteristics? | • Type or design • Affordability • Availability • Safety • Sustainability |
| 2. What is known about the ICS effectiveness? | • Black carbon • CO • $PM_{2.5}$ emissions • Health outcomes compared to the TSF |
| 3. What other measures support the implementation of the ICS? | • Education • Awareness campaigns • Incentives • Support |
| 4. What are the users' perceptions of the ICS identified from questions 1 and 2 above? | • Health • Family • Timesaving • Cooking and cultural practices • Fuel saving |

- *Time frame*: From August 2014 to September 2022 to include all ICS designs studied after the 2014 publication of WHO Indoor Air Quality Guidelines and interim $PM_{2.5}$ target for HAP.

- *Study type and publication*: Empirical qualitative, quantitative, or mixed-methods studies published in peer-reviewed journals and/or on relevant organisation websites.

- *Type of Intervention and outcome*: ICS intervention studies with personal and/or household measurements of HAP (CO, $PM_{2.5}$, Black Carbon) and/or reported health outcomes and/or user perspectives

- *Population*: Households in sSA.

- *Language*: Published or auto-translated into English language due to limited translation resources.

  We excluded sources if

- Studies were laboratory-based

- Interventions were LPG and electricity (target populations are unlikely to have access in the next 10–20 years)

- Interventions were based on solar or biogas

- There was no name or description of the improved cookstove

**3. Selection of evidence.** Search duplicates were removed from ENDNOTE® X9 reference manager software before exporting to Rayyan®, a platform that allows multiple collaborators to simultaneously screen and code the studies [38]. Four reviewers (EP, JL, MD, NK) conducted a three-stage blinded screening process using the established inclusion and exclusion criteria. The title and abstract screening preceded the full-text retrieval and screening. To ensure consistency, each reviewer double-screened a 20% random selection of each other's work. Consensus was achieved through group consultations and discussions [33]. The final data screening occurred during the data charting process.

**4. Data charting.** Seven reviewers (EP, JL, MD, NK, DS, AW, VJ) independently extracted data from included studies. Non-sSA sites and non-biomass cookstoves' data in our included studies were excluded at the data charting stage. In one instance, there were two papers reporting on the same study. We therefore only included the paper with the more comprehensive analysis. This paper reported the same data but included potential confounders in the analysis including fuel used for lighting, number of cooking episodes, and average number of people cooked for. We deemed this a more robust picture of reported pollutant levels in real-life settings.

We utilised a Microsoft Excel$^{©}$ spreadsheet to chart domains specified by the review questions. These domains included study design, location, duration of the intervention, population, ICS characteristics (design, fuel, combustion type), comparator (three-stone fire or traditional cookstove), ICS features (sustainability, safety, cost), supporting interventions (incentives and awareness), outcome measures ($PM_{2.5}$, CO, black carbon, health outcomes), and user perceptions (S2 and S3 Figs). In addition, where required, missing information, such as ICS name, description, cost, market availability, and tier ratings, was sought from corresponding authors, Clean Cooking Alliance Catalogue [36], and marketing websites.

*4.1. Quality appraisal.* Although an optional process in a scoping review [34], a quality appraisal was deemed appropriate to assess the quality of available ICS studies. The studies were appraised using the Liverpool Quality Assessment Tool (LQAT) [16] for quantitative

studies, the adapted version of Harden et al.'s [15] appraisal tool for qualitative studies, and the global rating scale [39]. Mixed-method studies were appraised with both appraisal tools. Assessment entailed rating the studies based on study context, methodology, baseline and outcome assessments, analysis/confounding, and impact of the findings to the review. Each element was assigned a rating of strong, moderate, or weak. The studies were rated as strong if there was no weak element, or moderate if one weak, and all other cases were rated as weak. if there was no weak. We provided an example, each of qualitative and quantitative study appraisal in S4A and S4B Fig.

**5. Summary synthesis and reporting results.** Validated conversion metrics for $PM_{2.5}$ [27] and CO emissions [40] were used to standardise emission units to facilitate comparison. The conversion was not done for g/kg of fuel weight as conversion could not be justified without parameters such as fuel water content. The resulting data were categorised by ICS type, draft system, fuel type, HAP, and health outcome measures. In addition, fuel type, chimney, draft, and type of combustion chamber were mapped with HAP emissions, health outcomes and, subsequently, against the ICS classification. We labelled kitchen and household HAP measurements as 'household' measurements due to the studies' limited descriptions of the term 'kitchen'. Furthermore, in this review, we labelled the comparators, i.e., traditional cookstoves and three-stone fire, as TCS/TSF.

*5.1 Post-synthesis screening of ICS to be promoted.* Following the synthesis of emission levels and health outcomes, the ICS were evaluated to identify which cookstoves should be promoted to poorer communities in sSA. Cookstoves were excluded if (1) there was no reduction in emission level or health improvement, (2) they were no longer available on the market irrespective of HAP reduction levels, (3) priced $\geq$ \$40 (mid-range of prices), (4) manufactured outside Africa (added shipping and import fees and difficulties in accessing customised parts reduce sustainability), (5) uses only pellets as fuel (the high price increases overall household cooking expenditure, and (6) uses charcoal fuel (ineffective use emits high PM, carbon monoxide, nitrogen oxides, sulphate oxides and other volatile compounds. These compounds have severe implications for health outcomes, e.g., cancer, low birth weight, and exposure associated with equivalent to smoking two packs of cigarettes/day [41]).

The user perspectives of included ICS were subsequently explored to inform their suitability to meet user needs. All findings are summarised in textual descriptions, data tables, and figures.

*5.2 Exploring the user perceptions.* We conducted the user perspective literature search in two phases. First, relevant studies were extracted from the initial search result. Second, a brand-specific search of ICS was executed in February 2021 and updated in July 2021 and September 2022 on PubMed, Global Health, Google Scholar, and relevant organisations' websites, including United Nations High Commission for Refugees. In addition, studies were included if conducted in sSA, published after 2014, and explored the experiences and user perceptions of the ICS that met review questions 1, 2, and 3. The findings are presented in descriptive and diagram format using relevant themes from existing literature [15, 42, 43].

## 6. Consultation with experts

We engaged with experienced researchers, social scientists, and an engineer, all with extensive experience in HAP, cookstove research and behavioural change approaches in LMICs to develop review questions, themes, methodology, and data extraction processes.

## Results

Overall, the review included a total of 33 studies, including 27 for review questions 1,2 and 3, and 10 that explored the ICS's user perspectives. All were peer-reviewed articles, except for two non-peer-reviewed publications [44, 45] which reported the user views of the *Save80*

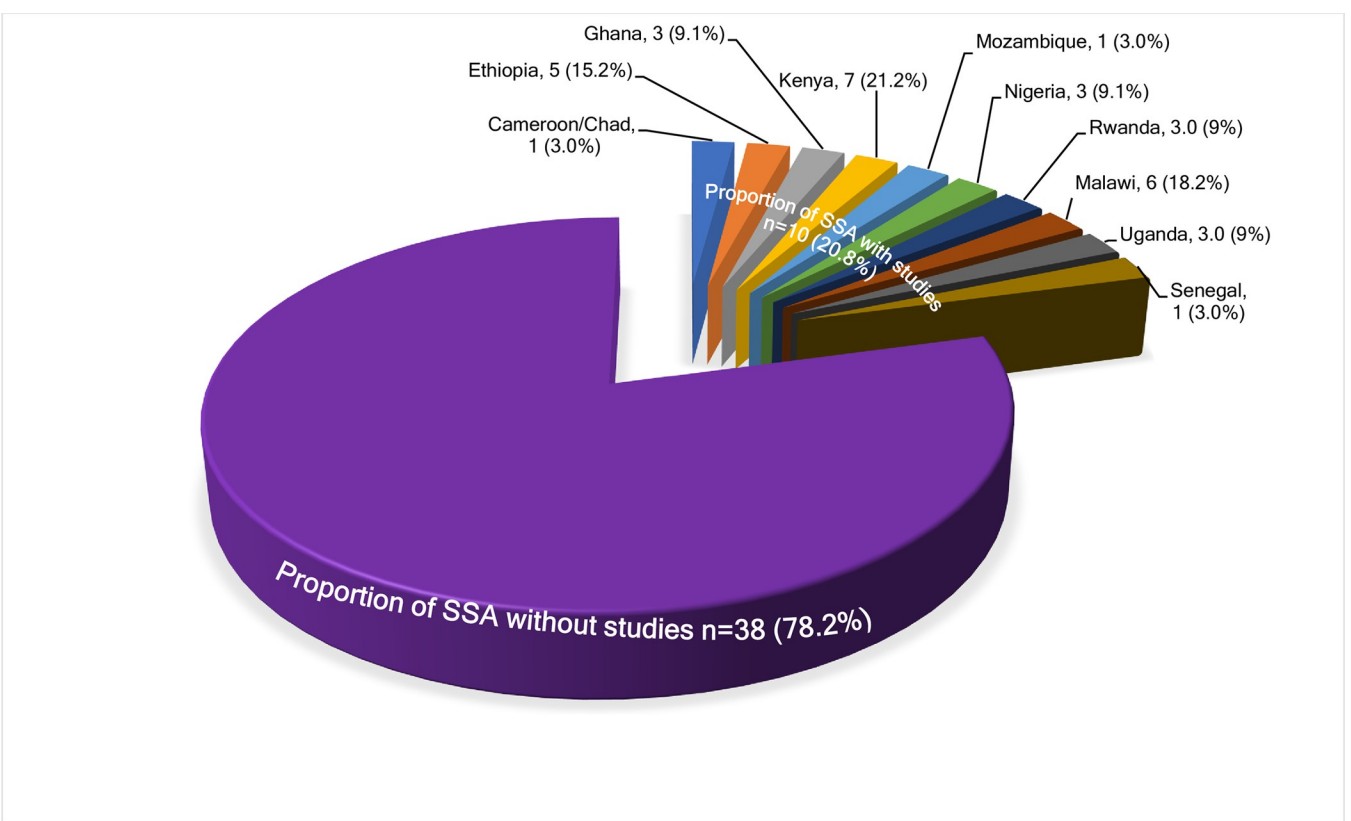

**Fig 1. Representation of the 33 reported studies across the sub-Saharan African countries.**

cookstove. Reviewed studies were conducted in only 10 of the 48 SSA countries [46] (Fig 1), spanning 35 study sites, 27 of which were in rural areas. Where sites were peri-urban and urban, studies [5, 47–49] described them as socio-economically disadvantaged areas.

The review included six clustered- and five randomised controlled trials (cRCT) and (RCT), respectively. Other designs included pre-post (n = 5), case-control (n = 2), comparative (n = 2), uncontrolled infield test (n = 3), controlled cooking (n = 2), quasi (n = 2), cross-sectional (n = 1), qualitative structured interview (n = 1), and two mixed-method studies. The ICS were comparable to the three-stone fire in all the studies. Only 10 of the 33 studies reported additional techniques to support the ICS interventions. These included awareness/education/training (n = 7), behavioural change (n = 1), participatory methods (n = 2), and subsidised repairs and maintenance (n = 3). However, none reported the impact of the measures on study outcomes. Our quality assessment of included studies scored 12 as strong, 17 as moderate, and four studies as weak quality.

The study characteristics of included studies are displayed in Table 2 below.

We divided our result sections to reflect the different review questions. Section A reports on review questions 1, 2, and 3 findings; and section B on question 4, the exploration of user perspectives of improved cookstoves.

## Section A: Characteristics of improved cookstoves, household air pollution and health outcomes

**Selection and characteristics of studies.** Of the 1905 records retrieved from electronic databases, relevant websites, and grey literature searches, we found 27 field studies with HAP

**Table 2. Characteristics of included studies.**

| # | Source | Country | Study design | Population studied | Study Sample size N (I; C) | Intervention | Control/ Comparator | Personal | Household | Follow-up time | Quality appraisal score |
|---|--------|---------|--------------|--------------------|-----------------------------|--------------|----------------------|----------|-----------|----------------|-------------------------|
| | | | | | | | | **Section A: Sources examining emission levels and health related outcomes of cookstoves** — Types of measurement | | | |
| 1 | Adane et al., 2021 [50] | Ethiopia rural | C-RCT | 2031 low-income HH from North-west Ethiopia | 1977 HH | Mirt | Traditional cookstove | Not measured | Baseline & follow-up indoor PM2.5 (1hr) | 12 months | Moderate |
| 2 | Adane et al., 2021 [51] | Ethiopia rural | C-RCT | <4year old children from low-income HH in North-West Ethiopia | 5508 children | Mirt | Traditional cookstove | Incidence of childhood acute lower respiratory infection (ALRI) | Not measured | 12 months | Moderate |
| 3 | Alexander et al., 2017 [5] | Nigeria urban | RCT, | Women who are HH primary cooks, <18 weeks pregnant and use wood/ or kerosene as primary cooking fuel. | 324 (162;162) pregnant women | Cleancook Chulika Smoke awareness campaign | Traditional cookstove/ kerosene or firewood | 3 consecutive blood pressure (SBP & DBP) measurements at 16,20,26,30, 34, & 38 wk. gestation | Not measured | 4.9 months | Moderate |
| 4 | Bensch and Peters 2015 [23] | Senegal rural | C-RCT | Women responsible for cooking in 253 HH in 12 villages | 253 (98;155) HH | Firewood Jambar | Three-stone stoves & traditional metal woodstove | Self-reported respiratory and eye problems | Not measured | 12 months | Strong |
| 5 | Champion & Grieshop 2019 [47] | Rwanda urban /peri-urban | Case-control | HHs in Gisenyi low-resourced communities using biomass and locally manufactured charcoal as fuel | 22HH (14 Mimi Moto; 4 TSF & 4 Charcoal stoves) | MimiMoto Educational program | Three stone fire and charcoal stoves | Not measured | Fuel-based in-use emission $PM_{2.5}$, CO, Black carbon organic and elemental carbon | 36 months (yearly interval. 2015–2017) | Moderate |
| 6 | Coffey et al., 2017 [20] | Ghana rural | In-field Uncontrolled test | A subset of the intervention arm (N not stated) from the 200 HH in the main study (RCT) from the Kessana-Nankana districts | Intervention arm (i) 2 stoves (Gyapa/ Gyapa) in 50HH (ii) 2 stoves (Philips/ Philips) in 50HH (iii) 2 stoves (Gyapa/ Philips) in 50HH | Gyapa woodstove Philips HD4012 | Three stone fire Coal pot | Not measured | Real-time cooking emission. CO, CO2, PM organic carbon, elemental carbon | 1.5 months | Strong |

*(Continued)*

**Table 2.** (Continued)

| # | Author | Country | Study design | Setting | Sample size | Intervention | Comparison | Health and non-health benefits | HH time usage Fuel consumption | Follow-up | Quality |
|---|---|---|---|---|---|---|---|---|---|---|---|
| 7 | Cundale et al., 2017 [25] | Malawi rural | Qualitative: Semi-structured interviews | 10 HH from 10 village clusters in Chilumba. Mostly rural fishing and farming communities. HHs were participants in the CAPs 2015 trial | 100HH (50I, 50C) 100 semi-structured interviews | Philips HD4012LS User training & maintenance support | Traditional three-stone fire | | HH time usage Fuel consumption | 36 months | Moderate |
| 8 | Dutta et al., 2021 [52] | Nigeria peri-urban | RCT | Women in the early second trimester of pregnancy | 324 | CleanCook (Sweden AB) | Firewood/ Kerosene | Mother: 72hr Personal $PM_{2.5}$* Fetus: growth trajectories using biparietal diameter, head & abdominal circumference femur length, and ultrasound-estimate fetal weight | Not measured | 5.2 months (average follow-up) | Moderate |
| 9 | Garland et al., 2017 [48] | Uganda peri-urban | Comparative study | 16 HH in community outside of Kampala | 16 HHs | Referred to as TEG rocket | Traditional three-stone fire and Charcoal stoves | Not measured | Real cooking time uncontrolled black carbon emission | < 1 month | Weak |
| | | Kenya, urban | | 22HH Urban community, Kwangware, Nairobi | 22 participants | Kenyan Ceramic Jiko | Traditional cookstove | | | | |
| 10 | Gebreegzrabher et al., 2018 [53] | Ethiopia, rural | Controlled cooking test | 108 HH from 81 villages in the forestry region | 108 HH from 360 HH that received the ICS | Mirt | Traditional three-tripod stove | Respiratory discomfort | Not measured | 5–6 months | Moderate |
| 11 | Gitau et al., 2019 [54] | Kenya rural | Quasi | HHs in Waa ward, Matunga constituency, Kwale | 25HH for ICS; 5HH for TSF (different cooking test dates) | Gastov | Three stone fire | Not measured | Real-time $PM_{2.5}$, CO, CO2 | 2 months | Strong |
| 12 | Hankey et al., 2015 [55] | Uganda rural | Pre-post design | 54 HH in 6 rural Ugandan villages surrounding Kyetume | 28 HH for $PM_{2.5}$ and 34 HH for CO. | Ugastove | Three stove fire | Not measured | 48 hr $PM_{2.5}$ 24 hr CO | 1 month | Moderate |
| 13 | Jagger et al., 2019 [49] | Rwanda urban | Cross-sectional (impact evaluation study) | 144 HH decision-makers (>15years) and primary cooks in 22 communities in Bugoyi and Kivumu of Gisenyi district who had never used an ICS. | 91 primary cooks at midline who were present at baseline. | Mimi Moto Marketing strategy Door-to-door visits Cooking demonstration | Traditional and charcoal stove | Blood pressure, Shortness of breath Cooking time | Not measured | 8 months | Moderate |

(Continued)

**Table 2.** (Continued)

| # | Author | Country | Study design | Participants | Sample size | Intervention | Comparison | Health outcomes | Real-time cooking measurement | Duration | Strength |
|---|--------|---------|--------------|--------------|-------------|--------------|------------|-----------------|-------------------------------|----------|----------|
| 14 | Jagoe et al., 2020 [56] | Kenya rural | Pre-post design (exploratory sequential) Mixed study | Participants who do most of the cooking in HHs from 3 rural agricultural communities (Githembe, Kambaa, and Bathi) | 55HH | Kuniokoa | Three stone fires | Not measured | Real-time cooking measurement using SUM | 3.5 months | Moderate |
| 15 | Jary et al, 2014 [24] | Malawi, rural | RCT (feasibility study) | Non-smoking women in the Ntcheu district who cooked on traditional open wood fires and wished to purchase a chitetezo stove | 51 (25;26) women | Chitetezo stove | Traditional open fire | Exhaled CO, shortness of breath, wheezing, eye problems, back pain, using questionnaire | 24 hr Ambient CO | 0.2 months | Moderate |
| 16 | Kirby et al., 2019 [57] | Rwanda, rural | C-RCT | Primary cook and children under 5 years in poor region of Ududehe region | 1582 HH (789, 793) HH | Eco Zoom Dura community and HH education, e.g., radio songs behavioural change messaging | Traditional biomass stoves | Acute respiratory infection in CU5 48 hours $PM_{2.5}$ | None | 12 months | Strong |
| 17 | LaFave et al., 2021 [58] | Ethiopia, rural | RCT- post-intervention | 480 HH in 36 communities | All children and adult cook | Mirt Stove Training and ICS awareness | Traditional cookstove | Child growth, respiratory conditions, activities of living | 72 hours $PM_{2.5}$ | 40 months | Strong |
| 18 | Mortimer et al. 2017 [59] | Malawi, rural | C-RCT | 8470 HH in 150 communities with at least one child aged below 5 years | 10750 (5400;5350) children | Philips HD4012LS Philips SA | Traditional open fire | Pneumonia, severe pneumonia/death in CU5 | Not measured | 26 months | Moderate |
| 19 | Njenga et al., 2016 [60] | Kenya, rural | Quasi | Women in selected HHs in rural Kibungu, Embu County | 5HH from 57 baseline(trial) participants | Gastov Hifadhi ICS Participatory approach | Three-stone fire & Improved Hifadhi stove | Not measured | $PM_{2.5}$, CO Cooking time, fuel savings | Not explicit | Strong |
| 20 | Ochieng et al., (2017) [61] | Kenya, rural | Uncontrolled pre-post comparison | HH in siaya country with primary cook having a child under 5, and exclusively using biomass | 48HH | Rocket mud stove | Pre-intervention use of three-stone fire | 8 hours personal CO exposure | 48 hours kitchen CO levels, $PM_{2.5}$ | 10 months | Moderate |
| 21 | Onyeneke et al., 2018 [22] | Nigeria, rural | Pre-post design (cross-sectional) | HH in 9 rural communities in Kaduna where most HH depend on fuelwood for cooking | 280HH (70 adopters;210 non-adopters) | Save80 Mass media exposure | Three stone fire | Exhaled CO Sore eyes Cold | Real-time cooking CO measurement Fuel use Time savings | >6 months | Moderate |

*(Continued)*

**Table 2.** (Continued)

| Source | Country | Study Design | Population studied | Intervention | Comparison | Outcome measure | Outcome measure | Duration | Appraisal |
|---|---|---|---|---|---|---|---|---|---|
| 22 Pilishvili et al., 2016 [21] | Kenya, rural | Uncontrolled pre-post comparison (Cross-over design) | HH in 2 villages in Nyanza province with women (15–49 years old) and with one or more children under five. | 45HH | Ecochula EcoZoom Envirofit Philips Prakti Rocket (TECA) | Traditional three-stone fire | 48-hour personal CO exposure | 48 hours CO and $PM_{2.5}$ | 0.5 months -2 wks. use with 1 wk. interval | Strong |
| 23 Quinn et al., 2017 [62] | Ghana, rural | RCT | Pregnant women in HH in Kintapo north & south districts enrolled in the GRAPHs study. | 44 Women | Biolite $LPG^2$ | Traditional cookstoves | 24-hour ambulatory Home $BP^2$ monitoring and Personal CO | Not measured | Not explicit | Strong |
| 24 Rosa et al. 2014 [63] | Rwanda, rural | C-RCT | Head of HH (>18years) in 566 HH in 3 villages, Nyarutovu and Kabuga and Rubona, | 126 (63,63) HH | EcoZoom Dura stove Program support/ periodical HH visit. One-to-one training & maintenance Community participation | Three stone fire | Not measured | 24-hour average $PM_{2.5}$ exposure in the main cooking area (both indoors & outdoors) | 5 months | Strong |
| 25 Saleh et al 2022 [64] | Malawi, rural | Mixed: quantitative before-after intervention study and qualitative observations /discussions | Residents of rural HH of the study location n = 300 households | 18 individuals | Chitetezo mbaula Community engagement | Not applicable | Personal 48hr $PM_{2.5}$ exposure | Not measured | 3 months | Moderate |
| 26 Vaccari, Vitali & Tudor 2017 [65] | Cameroon/ Chad, rural | Comparative study | Logone Valley, the border of Cameroon and Chad. | 3HH | CentraAfricain Ceramic $ICS^2$ Rice-husk $burner^2$ $LPG^2$ Solar $Cooker^2$ | Traditional three-stone fire | Not measured | 8- hour exposure to indoor CO | Not stated | Weak |
| 27 Wathore, Mortimer & Grieshop, 2017 [66] | Malawi rural | Uncontrolled pre-post design | Not described | 22 HH with 45 HH cooking sessions. Cooking time (49 mins 15–223 min) median | ACE 1 Philips HD4012LS Chitetezo Mbaula | Traditional three-stone or simple mud stoves | Not measured | Indoor emissions of CO and particulate light scattering (proxy for $PM_{2.5}$) | 2 months | Strong |

| Section B: Sources examining user perspectives | | | | | | |
|---|---|---|---|---|---|---|
| Source | Country | Study Design | Population studied | Intervention of Interest | Factors explored | Appraisal |
| 1 Beck 2015 [44] | Nakivale, Uganda, rural | Not applicable | Organisation news report | Save80 | Time- saving, safety, fuel-saving, poverty alleviation | Weak |

(*Continued*)

**Table 2.** (Continued)

| | | | | | | | |
|---|---|---|---|---|---|---|---|
| 2 | Bensch and Peters 2015 [23] | Senegal, rural | C-RCT | Women responsible for cooking in 253 HH in 12 villages | Firewood Jambar | Time-saving benefits; Stove stacking; Others | Strong |
| 3 | Dickinson et al., 2019 [67] | Ghana, rural, urban | Pre-post comparative study | 200 HH from 25 geographical clusters | Gyapa | Time-savings benefits; Safety; Smoke; Health benefits; Fuel-saving; Suitability for traditional cooking suitability; Durability | Moderate |
| 4 | Dresen et al., 2014 [68] | Ethiopia, rural | Pre-post HH survey in RCT | 148 HH (96 adopters; 52 non-users) in Kafa province, SW forest area in Ethiopia. 266 ICS users | Mirt | Smoke reduction, Fuel-saving, Burns, Traditional suitability | Strong |
| 5 | Gebreeziabher et al., 2018 [53] | Ethiopia, rural | Controlled cooking test using survey | 360 HH from 36 sites in three Ethiopian forestry regions. | Mirt | Fuel-saving, time-saving, Smoke, stove satisfaction | Moderate |
| 6 | Jagger and Jumbe, 2016 [69] | Malawi, rural | Pre-post discrete choice survey | 383 HH | Chitetezo Mbaula | Time-savings; Safety; Smoke; Price; Stove stacking, Health; Fuel-saving; Traditional suitability; Efficiency, Fuel processing; Durability; Others | Moderate |
| 7 | Jagoe et al., 2020 [56] | Kenya, rural | Pre-post design (exploratory sequential) | Participants who do most of the cooking in HHs from 3 rural agricultural communities | Kuniokoa | Timesaving; Safety; Smoke; Health benefits, Gender appreciation | Moderate |
| 8 | Ndunda, 2017 [45] | Kakuma, Kenya, rural | Not applicable | Organisation news report | Save80 | Health and wellbeing, time-saving, fuel saving | Weak |
| 9 | Pailman et al., 2018 [70] | Mozambique/ Malawi, rural | Exploratory- User-centred approach (mixed methods including survey) | 126 HHs across the four countries, South Africa, Mozambique, Malawi, and Kenya, in urban, rural, and peri-urban areas | Chitetezo Mbaula Gyapa, Kenyan Jiko | Smoke; Price; Fuel-saving; Durability; Safety; Others | Strong |
| 10 | Saleh et al., 2022 [64] | Malawi, rural | Mixed: quantitative before-after intervention study and qualitative observations /discussions | Residents of rural HH of the study location | Chitetezo mbaula | Price, Time-savings, Fuel-saving, Durability, Accessibility | Moderate |

Table 2 reference key

**1** = GRAPHs: Ghana randomised air pollution and health study; CRCT = clustered randomised controlled trial; RCT = Randomised controlled trials; HH = households; SUM: Stove use monitors; N = total number of samples; I- Intervention group number; C- Control/Comparator group where applicable; SBP- systolic blood pressure; DBP- diastolic blood pressure. This review defines the kitchen as any area or room where stoves are used in the household. **2** = Excluded from synthesis. LPG and solar (exclusion criteria), Ceramic ICS and rice husk burner were reported using apposite laboratory values (exclusion criteria); Home BP was measured with the use of LPG. * Although measured in the study, the values were not reported cleary to report their impact on the reported health outcomes.

emission measurements and health outcomes (Fig 2). While most (20) reported kitchen/household $PM_{2.5}$ levels, only one and two studies had data on black carbon and personal $PM_{2.5}$ data, respectively. Although Duttal et al. [52] reported higher levels of $PM_{2.5}$ in the control

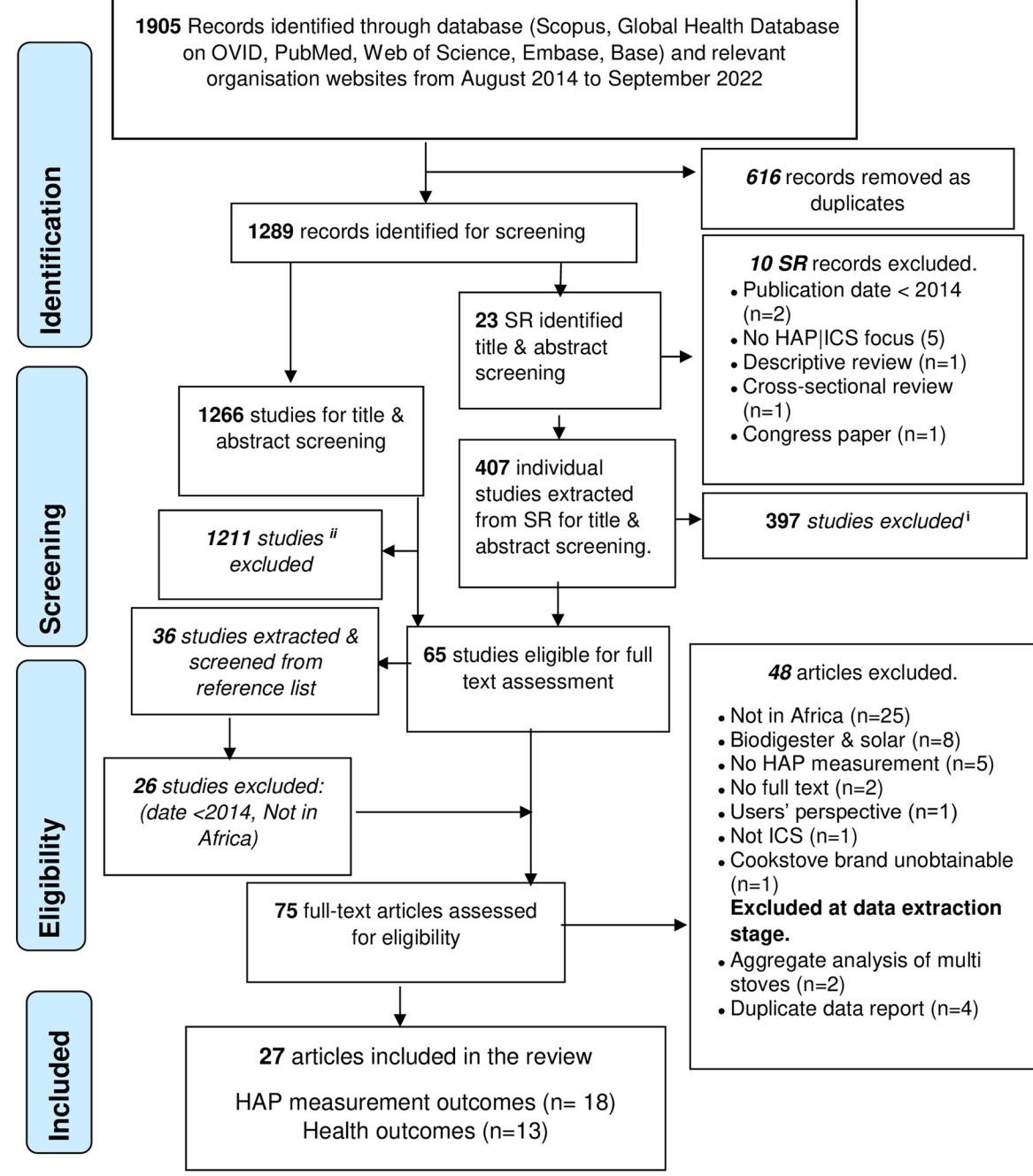

**Fig 2. PRISMA flow diagram of selection of studies included in HAP and health outcomes.**

group, the actual data was not reported and thus was excluded from personal $PM_{2.5}$ synthesis. Measured and self-reported health outcomes were available in 13 studies and included blood pressure (n = 2), burns (n = 5), eye problems (n = 4), fetal growth/weight (1), musculoskeletal issues (n = 2), pneumonia (n = 3), respiratory issues (n = 6), sneezing, cold and cough (n = 4). Fuel savings and cooking time were reported in 15 and nine studies, respectively. Only seven studies reported on cookstoves' durability. The follow-up period in these studies ranged from 0.2 [24] to 40 months [58], modal $\geq$ 6 months.

## Characteristics of improved cookstoves

We identified 23 ICS brands from the 39 tested cookstoves. The ICS were classified as improved/energy-efficient cookstoves (rocket style n = 13), advanced cookstoves (gasifier n = 3; semi-gasifier n = 1), and unclassified stove types (n = 6). The presence of a chimney (a feature that directs smoke away from the cooking area) was described only in *EcoZoom Dura* [57, 63], *Mirt* [53, 58], *Prakti Leo* [21], and *RTI-TECA* [21] studies. Other ICS either had no chimney (1/39) or was not reported (19/39) in the studies. The draft system (airflow system which impacts fuel burning rate, thermal efficiency, and completeness of combustion) [71] was described in only 11 cookstoves. Mostly (28/39), firewood was reported as the primary fuel. Other fuels included charcoal, pellet, and crop residues. Twenty-two cookstoves were locally made in sSA, with eight being produced industrially and four semi-industrial. For comparison, we defined 'locally made' as produced using local materials in sSA, 'industrially made' as imported as a whole unit, and semi-industrial as parts imported but cookstoves assembled locally.

**HAP outcome measures: Black carbon, PM2.5, and CO levels compared to TCS/TSF.** The household and personal HAP emission levels are summarised and presented with the cookstoves' characteristics and study design in Table 3 below.

Except for the increase in percentage reduction in household CO with *EcoZoom* [21] (1.96% n = 36), *Hifaldhi* and personal CO (1.45% n = 4) with *Biolite* [62], all reviewed ICS reduced HAP emissions compared to TCS/TSF although the amount varied across the different studies and between same brand cookstoves such as Gastov and Philips HD4012LS+ (Table 3).

The outcome of the percentage reductions in household-level $PM_{2.5}$ and CO and personal-level CO compared to TCS/TSF are displayed in Fig 3A–3C.

Surprisingly, ICS without chimneys showed the highest percentage reductions across the household and personal $PM_{2.5}$ and CO levels. However, the *EcoZoom* dura [63] with a chimney demonstrated a higher percentage reduction (-46%) in household $PM_{2.5}$ levels compared to *EcoZoom* [21] without a chimney (-20%) (Fig 3A).

All the ICS tested failed to reduce household $PM_{2.5}$ to a level close to the WHO 24-hour average of $0.025mg/m^3$ for safe indoor air quality in LMICs [29]. The lowest level of $0.11mg/m^3$ reported with the rocket-design EcoZoom [21] cookstove was over four times the WHO-IAQ-safe level. Measured in PPM, 75% of the 12 household CO levels measured were within WHO recommendations of 6.11PPM [29], with the lowest mean of 0.2PPM in the EcoZoom cookstove [21]. The available data on black carbon shows a mean level of 0.69 g/kg (n = 11) with the forced draft *thermoelectric generator* (TEG) [48] and 0.28g/kg (n = 32) with the *Kenyan Ceramic Jiko* cookstoves. Respectively, this represents a 38% increase and a 37% decrease compared to TSF and TCS. While the study [60] recorded this difference in terms of surface oxidation of the charcoal cookstove (a process where PM formation is less likely), the sample size of only 11 cookstoves with the TEG compared to 32 with the charcoal cookstove may be a factor in the precision of the TEG measurement. Also, the mean CO level was higher in the *TEG* (0.50g/kg) than the charcoal cookstove (0.44g/kg).

**Table 3. Summary of findings of HAP measurements by ICS design, draft system, and brand.**

| Stove design | Stove draft system | Stove brand | Study design | Combustion chamber/Fuel type | *Chimney features | *Place manufactured[i] | *Personal levels ^ = Statistical test (study#) | | *Household levels ^ = Statistical test (study#) | | Monitoring time* (hour) | *Fuel % reduction/kg | *Cooking time reduction min/day | Cost of cookstoves (USD$) |
|---|---|---|---|---|---|---|---|---|---|---|---|---|---|---|
| | | | | | | | PM$_{2.5}$ mean; (% reduction); p-value; (n) | CO (mean); % reduction (95%CI); p-value; (n) | PM$_{2.5}$ (mean); % reduction (95%CI); p-value; (n) | CO (mean); % reduction (95%CI); p-value; (n) | | Compared to TCS/TSF | | |
| Rocket | Forced draft | TEG2[ii] [48] | Comparative | Stainless steel/Wood | U | NO | NM | NM | Black Carbon: Mean 0.69 g/kg 38% (95%CI 0.23; 0.25,1.60); p = 0.10 (n = 11) | | U | -51.0% | NM | NO |
| Rocket | Natural draft | Chitetezo Mbaula [66] | Pre-post | Unspecified/Wood | N | Locally made | NM | NM | 6.8g/kg; -12.82% p = 0.347 (n = 16) | 106g/kg; 8.16%↓ p = 0.51 (n = 16) | ≥24 | -26.0% | NM | $ 1–2[iii] |
| | | Chitetezo [24] | RCT | Unspecified/wood | N | Locally made | NM | 0.5 PPM[iv] -33.3% p = 0.04 (n = 50) | NM | NM | <24 | NM | NM | $2[iii] |
| | | Chitetezo Mbaula [64] | Pre-post | Clay/Maize cobs, Firewood, Charcoal | N | Locally made | 0.0019 mg/m$^3$ (-16%) p = 0.71 | NM | NM | NM | >24 | NM | NM | $ 1–2.5[v] |
| | | EcoZoom Dura[vi] [63] | CRCT | Unspecified/Wood | Y | Industrial | NM | NM | 0.485mg/m$^3$, (0.04,2.28); -46.4% p = 0.005 (n = 60) | NM | ≥24 | NM | NM | $30-40[v] |
| | | EcoZoom Dura [57] | CRCT | Unspecified/Wood | Y | Industrial | 0.218 mg/m$^3$ -55.1% p = 0.49 (n = 183)[vii]; 0.224 mg/m$^3$ -3.0%; p = 0.13 (n = 84)[vii] | NM | NM | NM | ≥24 | NM | NM | $30-40[v] |
| | | Mirt Stove [58] | RCT follow up | Unspecified/Firewood | Y | Locally made | NM | NM | 0.135 mg/m3-10.4% p = 0.5 (n = 202) | NM | ≥24 | NM | NM | $10[viii] |
| | | Mirt Stove [50] | CRCT | Concrete & volcanic ash/ Briquettes, wood, crop residue, dung | N | Semi industrial | NM | NM | 0.340 mg/m$^3$ -58% from baseline p = 0.0001 | NM | <24 | NM | NM | $2.9 - $6.1[v] |
| Rocket | Unspecified | Gyapa [20] | RCT | Ceramic/Wood | U | Locally made | NM | NM | 2.6g/kg; -18% (-47,27) p = 0.36 (n = 18) | 58g/kg; -21% (-41,7) p = 0.12 (n = 18) | ≥24 | -10.0% | -5 | $7[iii] |

*(Continued)*

Table 3. (Continued)

| Stove design | Stove draft system | Stove brand | Study design | Combustion chamber/ Fuel type | *Chimney features | *Place manufactured[i] | *Personal levels ^ = Statistical test (study#) | | *Household levels ^ = Statistical test (study#) | | Monitoring time* (hour) | *Fuel % reduction/ kg | *Cooking time reduction min/day | Cost of cookstoves (USD$) |
|---|---|---|---|---|---|---|---|---|---|---|---|---|---|---|
| | | | | | | | PM$_{2.5}$ mean; (% reduction); p-value; (n) | CO (mean); % reduction (95%CI); p-value; (n) | PM$_{2.5}$ (mean); % reduction (95%CI); p-value; (n) | CO (mean); % reduction (95%CI); p-value; (n) | | Compared to TCS/TSF | Compared to TCS/TSF | |
| | | Envirofit [21] | Pre-post | Metal alloy/ Wood | U | Industrial | NM | 1.3 PPM -54% (-2.0 to -0.6); p<0.01 (n = 30) | 0.277mg/m$^3$, -35.6% (25.7,44.2) p<0.001 (n = 35) | 3.4PPM; -27.6% (16.6,37.2) p = 0.02 (n = 34) | ≥24 | -22.5% | -14 | $99.95[v] |
| | | EcoZoom [21] | Pre-post | Ceramic/ Wood | U | Industrial | NM | 0.7 PPM -32% (-1.1 to -0.3); p<0.01 (n = 31) | 0.109mg/ m$^3$, -19.7% (7.6,30.2) p = 0.12 (n = 37) | 0.2PPM; 1.9% (-12.6,14.6) p = 0.89 (n = 36) | ≥24 | -29.2% | -12 | $30-40[v] |
| | | Prakti-Leo [21] | Pre-post | Steel alloy/ Wood | Y | Industrial | NM | 0.9 PPM -45% (-1.4 to -0.4); p<0.01 (n = 32) | 0.118mg/ m$^3$, -38.6% (29.5,46.5) p < 0.001 (n = 39) | 0.7PPM; -32.3% (22.3,41.0) p<0.01 (n = 37) | ≥24 | -20.8% | 2 | 39[ix] |
| | | RTI-TECA[x] [21] | Pre-post | Brick and clay/ Wood | Y | Locally made | NM | 0.8 PPM -35% (-1.5 to -0.1); p = 0.03 (n = 31) | 0.215mg/ m$^3$, -31.9% (21.1,41.3) p<0.01 (n = 35) | 2.5PPM; -25.1% (13.2,35.3) p = 0.05 (n = 34) | ≥24 | -31.7% | -2 | $130[v] |
| | | Uga-stove [55] | Pre-post (observational) | Unspecified/ wood | U | Semi-industrial | NM | NM | 0.68mg/m$^3$; -37% (0.2, -1.2); p<0.01 (n = 28) | 1.4PPM -8% (-5.2, -7.9) p = 0.68 (n = 34) | ≥24 | NM | NM | $17[xi] |
| | | Rocket Mud-stove [61] | Pre-post (longitudinal) | Unspecified/ wood | N | Locally made | NM | 0.9 PPM 11.6% (-4.3 to 2.6); (n = 23) | 0.345; -13.1% (SD 0.273mg/ m$^3$); (n = 33) | 3.1 PPM -28.1% (-8.1 to 1.8); (n = 23) | ≥24 | -20% | -60 | $2-4[xii] |
| | | Save80 [22] | Pre-post (Case control) | Unspecified /Wood | N | Semi-industrial | NM | 9.8 PPM[xiii] 96.4% (SD 46.97); p <0.001 (n = 70) | NM | NM | <24 | -46.5% | -91 | $17-37[v] |

(Continued)

**Table 3.** (Continued)

| Stove design | Stove draft system | Stove brand | Study design | Combustion chamber/ Fuel type | *Chimney features | *Place manufactured[i] | *Personal levels ^ = Statistical test (study#) | | *Household levels ^ = Statistical test (study#) | | Monitoring time* (hour) | *Fuel % reduction/ kg | *Cooking time reduction min/day | Cost of cookstoves (USD$) |
|---|---|---|---|---|---|---|---|---|---|---|---|---|---|---|
| | | | | | | | PM₂.₅ mean; (% reduction); p-value; (n) | CO (mean); % reduction (95%CI); p-value; (n) | PM₂.₅ (mean); % reduction (95%CI); p-value; (n) | CO (mean); % reduction (95%CI); p-value; (n) | | Compared to TCS/TSF | | |
| **Gasifier** | Forced draft | EcoChula [21] | Pre-post | Ceramic | U | Industrial | NM | 1.7 PPM -68% (-2.6 to -0.8); p<0.01 (n=31) | 0.116mg/m3; -18.0% (5.1,29.2) p=0.18 (n=36) | 1.7PPM; -21.5% (9.1,32.2) p=0.10 (n=34) | ≥24 | -37.5% | -16 | $29-33[v] |
| | | Philips [21] | Pre-post | Ceramic | U | Industrial | NM | 0.6 PPM -29% (-1.0 to -0.2); p<0.01 (n=29) | 0.357mg/m3; -45.2% (36.6,52.6) p<0.001 (n=35) | 2.7PPM; -38.5% (28.9,46.7); p<0.01 (n=35) | ≥24 | -55.8% | -21 | $89[v] |
| | | Philips HD 4012LS [20] | RCT | Ceramic/Wood | U | Industrial | NM | NM | 2.5g/kg; -13% (0,28) p=0.04 (n=11) | 45g/kg; -46% (-65,-18) p<0.01 (n=11) | ≥24 | -50% | -22 | $89[v] |
| | | Philips HD 4012LS [20] | RCT | Ceramic/Charcoal | U | Industrial | NM | NM | 1.6g/kg; -58% (-90,81) p=0.04 (n=13) | 92g/kg; -77% (-92,-34) p<0.01 (n=13) | <24 | -30.0% | -5 | $89[v] |
| | | Philips HD 4012LS [66] | Pre-post | Unspecified/Wood | U | Industrial | NM | NM | 4.1g/kg ± 0.6SD -47% p<.005 (n=8) | 52g/kg; -45%; p<0.005 (n=8) | ≥24 | -51% | NM | $90[iii] |
| | Forced draft | ACE-1 [66] | Pre-post | Unspecified/Wood | U | Not Specified | NM | NM | 6.8 ± g/kg p=0.158 (n=8) | 60g/kg (30,75) -40% | ≥24 | -27.0% | NO | $90[iii] |
| **Gasifier** | Natural Draft/TLUD[xiv] | Gastov [60] | Quasi | Galvanised steel/ Grevillea pruning | N | Industrial | NM | NM | 0.3mg/m3; -89% p<0.05 | 18PPM (±6); -45% (n=25) observation from 5HH | <24 | -27%[xiv] | -9[xv] | $35[iii] |
| | | Gastov [54] | Quasi | Galvanised steel/ wood | N | Industrial | NM | NM | 0.187 ± 75 mg/m3- 41% (n=5HH) | 6 (± 3PPM) -57% | <24 | -18% | 17[xvi] | $35[iii] |
| **Semi-gasifier** | Forced draft | Mimi Moto [47] | Case-control | Unspecified/Pellets | U | Industrial | NM | NM | 0.4g/kg p<0.05 | 14 g/kg[xvii] -97% p<0.05 | <24 | NM | NM | $40-65[v] |

(*Continued*)

**Table 3.** (Continued)

| Stove design | Stove draft system | Stove brand | Study design | Combustion chamber/ Fuel type | *Chimney features | *Place manufactured^i | *Personal levels ^ = Statistical test (study#) | | *Household levels ^ = Statistical test (study#) | | Monitoring time* (hour) | *Fuel % reduction/ kg | *Cooking time reduction min/day | Cost of cookstoves (USD$) |
|---|---|---|---|---|---|---|---|---|---|---|---|---|---|---|
| | | | | | | | PM2.5 mean; (% reduction); (n) | CO (mean); % reduction (95%CI); p-value; (n) | PM2.5 (mean); % reduction (95%CI); p-value; (n) | CO (mean); % reduction (95%CI); p-value; (n) | | | Compared to TCS/TSF | |
| Unclassified | Unspecified | Centra Africain [65] | Comparative | Unspecified /Wood | N | Locally made | NM | NM | Not measured | 11.96 PPM; -25.14% | ≥24 | -52.2% | -12 | $7 iii |
| | | Kenya ceramic Jiko [48] | Comparative study | Stainless steel/ Charcoal | N | Not specified | NM | NM | Black Carbon: 0.28 g/kg -36.6% (95%CI = 0.049; 0.07,0.63); p = 0.081 (n = 32) | NM | <24 | -45.0% | NM | $35-40 v |
| | | BioLite [62] | RCT | Unspecified | U | Industrial | NM | 1.45 PPM; 1.4% (n = 4) | NM | NM | ≥24 | NM | NM | $40-70 v |
| | | Clean-Cook Sweden AB [52] | Pre-post | Metal / Ethanol | N | Industrial | 0.023 mg/ m3 12.3% | NM | NM | NM | >24 | NM | NM | NO |
| | | Hifadhi Stove [60] | Quasi | Galvanised steel/ Grevillea pruning | N | Locally made | NM | NM | 4.25mg/m3 | 40PPM 11.11% | ≥24 | NM | 5 | NO |

Table 3 reference key

*: **U** = unspecified in the study; **NM** = not measured; **Y** = yes; **N** = no; **NM** = not measured in the study; **NO** = not obtainable; **n** = sample size; **PPM** = parts per million- the mass of a chemical or contaminate per unit volume of water.

^ = Unpaired student t-test (48); Two-sample test (66); Mann Whitney U-test (24, 52); Wilcoxon signed-rank test (64); Wilcoxon rank sum (64); Two-sided test (57, 58); Wald CI test (50); Paired t-test (21, 55, 61, 54); Non- parametric Kruskale Wallis test (60); Multi-level regression models (20, 22, 62); not explicit in the study (65). **i:** Semi-industrial described as domestically manufactured, or parts imported but assembled locally to enhance skill; locally made = made by local artisans in sSA, Industrial = manufactured and imported. **ii: TEG** = Thermo-Electric Generator cookstove. **iii:** Price obtained from the article. **iv:** Carbon monoxide was measured during participants' exhalation. **v:** Price obtained from the clean cooking catalogue. http://catalog.cleancookstoves.org/stoves. Accessed & updated October 5th, 2020 & February 3rd, 2022, respectively. **vi:** Two additional components, a "stick support" onto which fuel wood is placed to promote airflow and a "pot skirt" which increases fuel efficiency, were added to the stove in this study. **vii:** Outcome on primary cook (n = 183); Outcome in children under 5 years (n = 84). **viii:** Obtained from energypedia.info https://energypedia.info/images/a/a0/GIZ_HERA_2012_Mirt_stove.pdf Accessed February 3rd, 2022. **ix:** Price obtained from Engineering for change https://www.engineeringforchange.org/solutions/product/prakti-single-burner-wood-stove/ Accessed October 21st,2021. **x:** Built-in rocket stove with Thermoelectric-Enhanced Cookstove Add-on (TECA). **xi:** Price obtained from the article and www.ugastove.net. (subsidised for $7 for the study). **xii:** Single pot design. Price obtained from https://energypedia.info/images/1/11/GIZ_ HERA_2011_Shielded-fire-stove-with-bypass-air-inlet_Uganda.pdf Accessed October 5th, 2021 & updated Feb 3rd 2022. **xiii:** Endline mean value was not provided in the study. We calculated from baseline (187.9PPM), and estimated difference value (178.1PPM) described in the study as the average change in total household carbon monoxide exposure brought about by the adoption of Save80 cook-stove pgs 1331 &1333. **xiv: TLUD:** Top Lit Up Draft: 27% reduction in fuel reports is without the use of charcoal produced from the gasifier. The study reported the reduction increasing to 40% when charcoal is used. **xv:** The difference in the time taken to cook ugali and Sukuma meal using grevillea pruning in both the gasifier and TSF. **xvi:** The study reported the increase as the result of time to load, reload and time to light the fuel load needed for the gasifier. Cooking time is only lower by 1 min in the gasifier. Increase of 32% fuel saving if the char produced is used. **xvii** = Values obtained from the plot chart provided in the study. Data table or sources not available in the study.

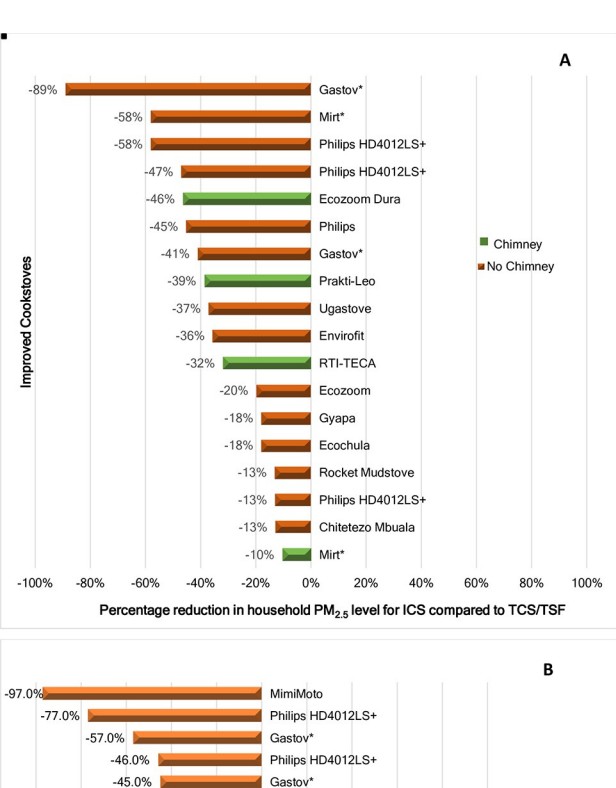

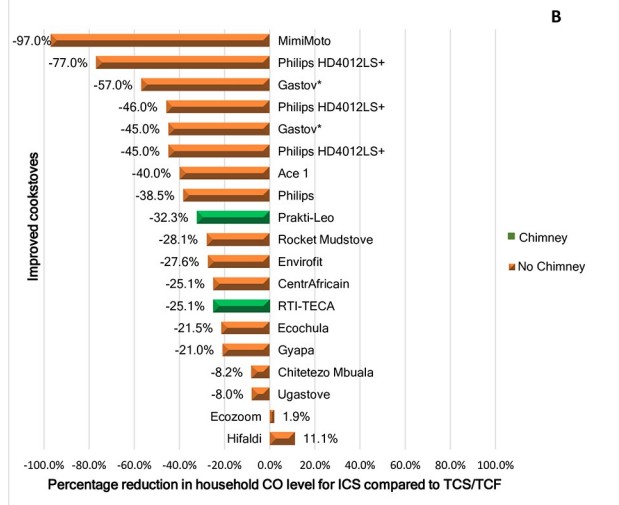

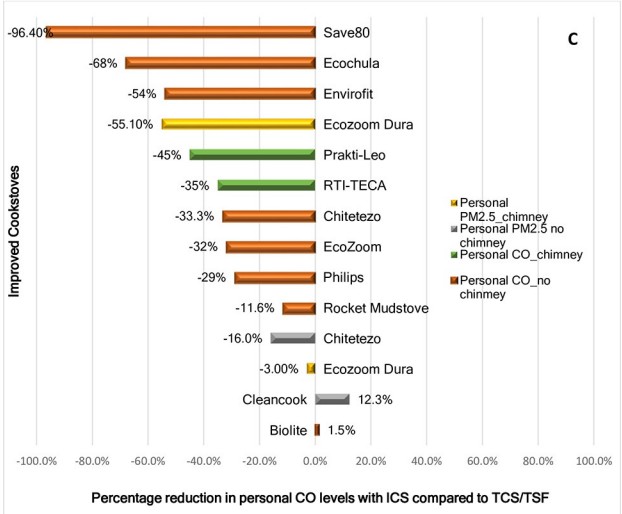

**Fig 3. Percentage reduction change in household and personal emission levels by ICS brand and chimney features.** (A) Bar chart shows percentage reduction in household PM2.5 with ICS brand compare to TCS/TSF (B) Percentage reduction in household CO measurements with ICS compared to TCS/TSF, variance in changes within same ICS brands, and % increase in EcoZoom and Hifadhi ICS (C) Personal CO and PM2.5 percentage reduction with ICS compared to TCS/TSF and a 1.5% and 12.3% increases in the Biolite and Cleancook respectively.

In addition, we found a large variation in the units used to report pollution levels. For example, household $PM_{2.5}$ and CO levels are reported in units of g/kg, $mg/m^3$ and g/kg, PPM, respectively, which creates difficulties in comparing HAP levels between studies. Also, some studies measure household CO (n = 19) and $PM_{2.5}$ (n = 21), whereas others measure personal CO (n = 9) and $PM_{2.5}$ (n = 3) and only reported two black carbon measurements. Table 4 shows the different units of reporting used in the reviewed studies by cookstove design and draft systems.

**Health-related outcomes.** A summary of health outcomes from the 13 studies where they were reported is presented in **Table 5**. The results were self-reported except for blood pressure [5, 49], fetal biometric parameters [52], and childhood pneumonia and incidence of ALRI [51, 57, 59].

*Blood pressure (BP).* Post-intervention BP levels showed a reduction in the two studies that measured BP. Respectively, the use of *Cleancook Chulika* [5] and *MimiMoto* [49] cookstoves reduced systolic blood pressure (SBP) by 0.60mmHg, and 3.32mmHg and diastolic blood pressure by 1.5mmHg and 2.37mmHg.

*Pneumonia.* The association between pneumonia and HAP was examined in three clustered randomised trials (cRCT) [51, 57, 59] and one semi-structured interview study [25]. There was no significant difference between the intervention and control group in the diagnosis of childhood pneumonia (incidence rate ratio [IRR]1.05, 95%CI 0.93–1.18, p = 0.44, n = 10471) and severe pneumonia (1.30, 0.99–1.71,p = 0.06) with *Philips HD4012LS and Philips SA* [59], prevalence ratio (PR) 0.87, 95% CI 0.58–1.30, *p* = 0.491, n = 5403 and severe pneumonia (PR = 0.75, 0.45–1.24, p = 0.256, n = 5403) with *EcoZoom Dura* [57], and odds ratio (OR) 0.95 (95% CI: 0.89–1.02, n = 5333) with the Mirt [51] cookstoves. The number of pneumonia cases was slightly higher in the intervention (n = 1255) than the control (n = 1251) group with the Philips brand [59] but slightly lower in the intervention group (41/2574) compared to the control (55/2829) with *EcoZoom Dura* [57]. The non-significant effects from the large sample could suggest that there weren't any meaningful changes in the groups over the short follow-up times

**Table 4. A snapshot of heterogeneous characteristics, units of measurement and range of HAP emission by ICS.**

| Draft system | ICS classification | Black Carbon (n) | Personal PM2.5 range (n) | Personal CO range (n) | Household PM2.5 range (n) | | Household CO range (n) | |
|---|---|---|---|---|---|---|---|---|
| | | g/kg | mg/m³ | PPM | g/kg | mg/m³ | g/kg | PPM |
| Forced | Rocket | 0.69 (1) | X | X | X | X | X | X |
| | Gasifier | X | X | 0.60–1.70 (2) | 1.60–6.80 (4) | 0.12–0.36 (2) | 45–92 (4) | 1.70–2.70 (2) |
| | Semi-gasifier | X | X | X | 0.40 (1) | X | 14 (1) | X |
| Natural | Rocket | X | 0.002–0.22 (2) | 0.50 (1) | 6.80 (1) | 0.14–0.49 (2) | 106 (1) | X |
| | Gasifier | X | X | X | X | 0.19–0.30 (2) | X | 6.00–18.00 (2) |
| Unspecified | Rocket | X | X | 0.70–9.8 (6) | 2.60 (1) | 0.11–0.68 (6) | 58 (1) | 0.20–3.40 (6) |
| | Unclassified | 0.28 (1) | X | 1.45 (1) | X | 4.25 (1) | X | 11.96–40.00 (2) |

involved. The qualitative study [25], which explored the views of 50 participants in the intervention arm (*Philips HD4012LS)* of the above cRCT study [59], found only four of the 50 participants self-reported reduced pneumonia incidence as an advantage and health benefit of the ICS.

*Eye symptoms.* All four studies measuring eye symptoms reported reduced eye discomfort, including eye pain, discomfort, and burning. Compared with pre-intervention data, self-reported reduction in eye-related symptoms was 70.4% (n = 778,p = 0.01) with *Firewood Jambar* [23], 66.7% (n = 25) with *Chitetezo* [24], 45.3% (n = 70) *with Save80* [22], and 20% (n = 50) with *Philips* ICS.

*Respiratory symptoms.* Five of the six studies that measured respiratory symptoms (defined in this review as the presence of cough, sneezing, wheezing, and difficulty breathing) reported a reduction in symptoms. Reductions were reported with the *Firewood Jambar* [23],

**Table 5. Description of health outcomes by author, ICS design and brand.**

| First author, date/ study design | [1]Name; Design; Combustion chamber; Fuel; Chimney | Description of participants | Health-related outcome change ^ = Statistical test (study #) | Burns/ Safety | Stove stacking | Fuel collection time min/ wk reduction | Cooking time minutes (% reduction) | Fuel savings | Sustain ability | Cost (USD) |
|---|---|---|---|---|---|---|---|---|---|---|
| Adane, 2021 [51] RCT | Mirt[a]; Rocket; Natural draft; wood No chimney | Low-income rural community of the Mecha Health and Demographic Surveillance System site. 5333 (I,2659; C, 2674) children <4 years | No evidence of reduced risk of childhood ALRI with intervention (OR 0.95 (95% CI:0.89–1.02). No statistically significant difference between the I and C group. | 20% reduced risk in children | × | × | × | × | × | $2.90– 6.10[2] |
| Alexander 2017 [5]/ RCT | Cleancook Chulika[a]; Rocket; Unspecified; Wood & ethanol No Chimney | 101 of the 324 pregnant women. (I: (Firewood to ethanol cookstove)[3] n = 50; C: firewood) n = 51. Pre (16wks gestation), Post (38wks gestation) | SBP: -0.60mm/ Hg (mean difference) DBP: -1.5mm/ Hg (mean difference) | × | Not reported in the group randomized to firewood. | × | × | × | × | $55 −80[2] |
| Dutta et al 2021 [52] RCT | Cleancook[a] Rocket; Unspecified; Wood & ethanol; Unspecified | 306 women in the second pregnancy trimester (C: (Firewood n = 152; I: ethanol cookstove)[3] (n = 15) in Ibadan, Nigeria, using $PM_{2.5}$ levels and fetal ultrasound measurement | No significant difference in growth trajectories between I & C group No sig association between $PM_{2.5}$ levels and fetal biometric parameters and intrauterine growth | x | x | x | x | x | x | $55 −80[2] |

*(Continued)*

**Table 5.** (Continued)

| First author, date/ study design | [1]Name; Design; Combustion chamber; Fuel; Chimney | Description of participants | Health-related outcome change ^ = Statistical test (study #) | Burns/ Safety | Stove stacking | Fuel collection time min/ wk reduction | Cooking time minutes (% reduction) | Fuel savings | Sustain ability | Cost (USD) |
|---|---|---|---|---|---|---|---|---|---|---|
| Bensch 2015 [23]/RCT | Firewood Jambaar[a]; Unclassified; Unspecified; Clay/Wood; No Chimney | Women responsible for cooking in 253 HH in 12 villages in rural Senegal. (I): n = 778; ©: n = 1199 | **Eye problems (Mean)** Prim cook (I): 2.9% & (C) 9.8%. % reduction = -70.4% n = 778 p = 0.01 **Respiratory Problems (Mean)** Prim cook: (I) 4.7% & (C) 11.8%. % reduction = -60.2% n = 778 obs. Male: 4.5 (-62%) p = 0.01 | × | × | 153 min/ wk. (mean diff) -15.49% | 84 min/day 20% reduction in cooking time | 27.7 mean difference = $ 2.03$ per month savings | 49% in use after 3.5 years | $10[4] |
| Cundale 2017 [25]/ Semi-structured interviews | Philips HD4012 LS[c]; Gasifier; Forced draft; Firewood; No chimney | 10 HHs from 10 village clusters in the Chilumba district. Primarily rural fishing and farming communities. HHs were participants in the CAPs 2015 trial. | Cough: (I): 1/ 50 Less smoke: (I): 10/50 (associated with reduced illness by respondent Less eye pain: 5/50 Reduced Pneumonia 4/ 50 (I); 6/50 (C) Reduced sneezing (C) 1 in 5 found no health benefits | × | × | 168min/ week (mean diff) -37.1% | 110 min/ day (mean diff) -50.2% | (I): 43/50; (C): 21/50 | I: 3/50. The solar panel was not durable. | $89[2] |
| Gebreegzrabher 2018 [53]/ RCT | Mirt[a]; Rocket; Natural draft; Firewood/ dung; Chimney | 360 treatment HHs | Self- Reported Less smoke Less respiratory discomfort | x | 88% stove-stacked with 3SF | x | 75% reported cooking time savings | -22% to -31% reduction compared to 3SF | x | $3.5-$7.3[5] |
| Jagger 2019 [49]/ RCT | MimiMoto[c]; Semi-gasifier; TLUD forced draft; wood pellets; No chimney | 91 primary cooks at midline (adopters) who were present at baseline from the 144 HH. (HH fixed effect) | SBP: -3.32mmHg (5.21) p <0.1 DBP: -2.37 mmHg (2.24) p = not significant SOB: -1.80 (0.86) p <0.01 48.8 (I) 41.06 (C): n = 182 | -1.64 (0.96) p <0.1 (self-reported) | Increased mostly during large cooking events | x | Mean reduction 0.7 ICS adopter's vs 1.9 non-adopters | No statistical significance between adoption of ICS and TSF fuel expenditure | x | $40 −65[2] |

*(Continued)*

**Table 5.** (Continued)

| First author, date/ study design | [1]Name; Design; Combustion chamber; Fuel; Chimney | Description of participants | Health-related outcome change ^ = Statistical test (study #) | Burns/ Safety | Stove stacking | Fuel collection time min/ wk reduction | Cooking time minutes (% reduction) | Fuel savings | Sustain ability | Cost (USD) |
|---|---|---|---|---|---|---|---|---|---|---|
| Jagoe 2020 [56]/ Pre-post exploratory sequential | Kuniokoa[b]; Rocket; Unspecified; wood; No chimney | 55 HH with participants who do most of the HH cooking in 3 rural agricultural communities | Self- Reported: Decrease smoke inhalation. Decrease intense heat. Less back strain from bending to blow on fire. ≈ 8hrs/wk of increase sleep time | Reduced risk of burns & intense heat. Fewer safety concerns | Increased by 75% (n = 41) from n = 5 baseline. ICS use 93 vs 267 min/day TSF | Reduced 414 min/ wk. (mean diff) @14 wks. -58.48% | 69 min/day (mean diff) -19.49% | x | x | $38[6] |
| Jary 2014 [24] RCT- feasibility | Chitetezo[a] Rocket; natural draft Wood No Chimney | 51 non-smoking women in rural Malawi who cook primarily on a TSF and wants to purchase chitetezo n: I = 25; C26. | Cough:28.57 Back pain 60% SOB 100% Eyes burning: 66.67% Sneezing & running nose: 150 increase cases | Increase cases in both groups. | × | × | × | × | × | $2[2] |
| Kirby 2019 [57] cRCT | EcoZoom dura[c] Rocket; Natural draft; Wood; Chimney | 793 HH- control & 789 HH- intervention of ARI in CU5 in poor region of Ubudehe | Current pneumonia: 41/ 2574 compared to 55/2829 control group p>0.05 | 1.8 vs 3.6 (rate)cases in control. P<0.001 | × | × | × | × | × | $30 −40[2] |
| LaFave 2021 [58] RCT- post intervention | Mirt Stove[a] Rocket; Natural draft; Unspecified; Chimney | All children and adult cooks from 480 HH in 36 communities in rural Ethiopia | Child growth: 0.06 SD taller than in control (not significant) Respiratory- Adult: No significant difference Respiratory- Child: Reduced in children ≤5yrs (p = 0.14) but not in older children Activities of daily living- Primary cook: minimal difference, not significant | × | × | × | × | × | 60% of ICS in use at 40 months post- intervention | $10[4] |
| Mortimer 2017 [59] cRCT | Philips HD4012LS[c]; Philips SA[C] Gasifier; Forced draft; Unspecified; No chimney | 8470 HH in 150 communities with at least 1 child under 5 in rural Malawi- CAPs. I: n = 1255, C: 1251 | Childhood Pneumonia: IRR (95% CI) p. IMCI[7] 1.05 (0.93–1.18)0.44 All pneumonia cases 1.02 (0.91–1.13)0.75 no evidence of association | 10% reduction in serious cooking- related burns & 42% in non- serious burns | Reported as a possibility with a high rate of solar panel and cookstove breakdowns | × | × | × | High rate of breakdown of solar charging panel cookstove | $89[2] |

*(Continued)*

**Table 5.** (Continued)

| First author, date/ study design | [1]Name; Design; Combustion chamber; Fuel; Chimney | Description of participants | Health-related outcome change ^ = Statistical test (study #) | Burns/ Safety | Stove stacking | Fuel collection time min/ wk reduction | Cooking time minutes (% reduction) | Fuel savings | Sustain ability | Cost (USD) |
|---|---|---|---|---|---|---|---|---|---|---|
| Onyeneke 2018 [22] Pre-post | Save80[b] Rockets; Unspecified; Wood; No chimney | 280HH (70 adopters; 210 non-adopters) in 9 rural communities in Kaduna with high reliance on firewood for cooking | Cough 3.38 (17.77%) reduction in cases Sore eyes 3.72 (- 45.26% reduction | × | × | 614 min/ wk. (mean diff) -46.51% | 91 min/day (mean diff) -38.32%[v] | 5.671kg/wk adopters' vs 0.002 kg/ week (non-adopters) 80.58% | × | $20 −55[2] |

Table 5 reference key

Abbreviations: TLUD: Top Lit Up Draft. I = Intervention Group; C = Control Group; n = number of observations; HH = households; ALRI = acute lower respiratory infections; SOB = Shortness of Breath; CAPs = Cooking and Pneumonia Study;; SBP = Systolic Blood Pressure; DBP = Diastolic Blood Pressure; IRR = Incidence Rate Ratio; OR = ODD ratio; SD = Standard Deviation; Prim cook = Primary cook in the sampled house.

^ = Two-sided test (51, 23, 57, 58); Fisher's exact test (5); Breusch-Pagan and Hausman tests (49); Paired t-test (56); Cox regression (59); Multi-level regression models (22).

1. a = Locally made; b = Semi-Industrial; c: Industrial; Semi-industrial is described as domestically manufactured, or parts imported but assembled locally to enhance skill acquisition, and locally made as produced locally using local materials in this study. Unclassified = Unclassified stove design in the study and no information on the web. Unspecified = Unspecified draft or chimney system or fuel used in the study; **2**. Price obtained from clean cooking catalogue. http://catalog.cleancookstoves.org/stoves. Accessed & updated October 8th, 2020 & February 3rd, 2022, respectively; **3**. We reported only outcomes from the participant group that compared ethanol cookstoves to firewood, i.e., excluded comparison to kerosene users, as this is not part of this review's objectives; **4** Price obtained from the article; **5**. Obtained from energypedia.info https://energypedia.info/images/a/a0/GIZ_HERA_2012_Mirt_stove.pdf; **6** Price obtained from https://www.burndesignlab.org/projects/kuniokoa# Accessed February 3rd, 2022; **7** IMCI = WHO Integrated Management of Childhood Illness, -defined pneumonia episodes in children under 5 years (CU5) of age diagnosed by physicians, medical officers, or other appropriately trained staff at local health-care facilities routinely accessed by trial participants, unaware of intervention allocation [54]

MimiMoto [49], Mirt [53], Philips HD4012 [25], Save80 [22], and the Chitetezo [24] cookstoves. In addition, a decrease in complaints of shortness of breath was statistically significant among adopters of MimiMoto [49] (p = 0.01, n = 182). While all participants (n = 25) with the Chitetezo reported reduced shortness of breath, increased sneezing or running nose was mentioned in 20.8% at follow-up compared to baseline (8.3%) [24].

*Burns- a proxy for safety*. Of the 13 studies on ICS-related health outcomes, data on burns were reported in only six, with 78% reporting a reduction in cases with the ICS intervention. Reduction in prevalence of burns between intervention and control groups was significant with EcoZoom Dura [57] (PR 0.51, 95% CI 0.36–0.74, p < 0.001), Philips HD4012 [59] (IRR 0·58 [95% CI 0.51–0·65]; p<0·0001). In self-reported cases, severe burns (including death) were reduced by 10% (n = 19) and in non-severe burns by 42% (n = 1505) with Philips HD4012. In addition, participants reported fewer safety concerns and reduced risks of burns with MimiMoto [49] and Kuniokoa cookstove [56]. In the locally made stoves, 41 cooking-related burns with Mirt compared to 51 cases in the control group [51]. However, burns incidence did increase in the intervention and control groups of the Chitetezo [24].

*Gender-specific health outcomes*. Associated HAP disease incidence and gender roles were highlighted only in the Bensch et al. study [23]. Though there was a substantial decrease in respiratory diseases and eye problems in users of Firewood Jambar compared to the control

group, the incidence of respiratory disease was 1.2 times higher in women who cook than in men within the same households despite the intervention.

*Other health-related outcomes.* Six of the studies reported other noteworthy health-related outcomes, including a self-reported reduction in back pain and strain in 60% (n = 25) of *Chitetezo* users [24], reduced back pain, increased sleep time of up to eight hours/week with the *Kuniokoa* [56], and reduced level of smoke with users of the *Kuniokoa* [56], *Mirt* [53], *and Philips HD4012* [25] cookstoves. The study on perinatal health using the *Cleancook* ethanol stove showed no significant association between exposure to PM$_{2.5}$ and fetal biometric parameters and similar fetal growth trajectories in the intervention and control groups [52].

**Availability of cookstoves.** We obtained information on market availability for 17 out of the 23 cookstove brands, including locally made (n = 6/23), semi-industrial (n = 3/23), imported (n = 7/23), and the *Philips* stove, which is no longer manufactured (n = 1/24) ICS. The six cookstoves (*CentrAfrican, Gastov, Rocket-Mudstove, RTI-TECA, TEG, and Ugastove*) with missing information were classified as unavailable in this review. We compiled and presented the descriptions of the available cookstoves in the S2 Table.

**Affordability of cookstoves.** The market price was available for 21 of the 23 cookstoves brands. The available prices ranged from $1–2 for the locally made *Chitetezo* [24, 66] to $130 [21] for th*e* industrially made *RTI-TECA*. Fig 4 below shows that only 25% of stoves cost ≤$10, half ≤$35, and three-quarters ≤$55. A cost beyond the budget of most households in poor communities in sSA. An average price was calculated when stove prices varied with the same brand.

**Sustainability.** Information on the cookstoves' sustainability (defined as evidence of stove breakdown and repair needed) was available in only four of the 27 studies. The *Mirt* [58] and the *EcoZoom Dura* [63] studies report 60% (n = 480HH) continuing functionality at 40 months

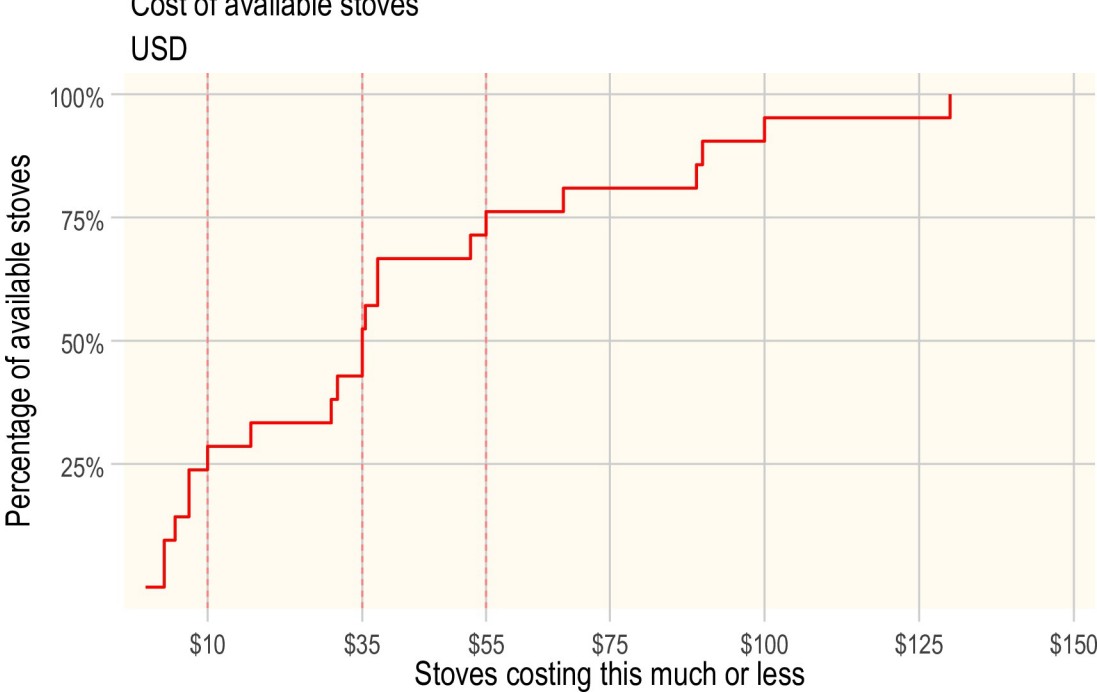

**Fig 4. The cummulative distribution chart of the stepped rises in the number of cookstoves (n = 21) available at or below the given cost.**

and 5% (n = 60) required repairs at five months follow-up, respectively. Similarly, almost half of the 253 HH users of the *Firewood Jambar* [23] still used the ICS at 3.5 years post-intervention, with technical problems reported as rare. In contrast, the *Philips* [59] and the *Gastov* [60] users reported a high number of stove breakdowns with limited availability of repair parts. Participants ranked the durability of the CentrAfrican ICS as similar to the TSF [65].

**Stove stacking.**   Evidence of stove stacking (concurrent use of multiple cookstoves) was reported in seven of the 27 HAP and/or health outcome studies with varying justification for use and comprehensiveness of reporting. Stove stacking was reported as being due to ICS 'not being suitable for households with greater cooking needs' with *MimiMoto* [49] and being 'unsuitable for baking local dishes' with the Mirt [58] cookstoves. Overall $PM_{2.5}$ reduction was unsurprisingly greater in households without stove stacking' [21]. Although Jagoe et al. [56] distributed multiple Kuniokoa cookstoves to households, the study observed stove stacking with TSF, although no information was provided on users' justification by the study.

**Fuel savings.**   Compared to inefficient cookstoves, the reduction in the fuel used with the ICS varied among the 15 studies that measured fuel savings. The reduction in firewood use ranged from 10% with the locally made *Gyapa* [20] to 80.6% with the semi-industrial *Save80* [22]. When reported in terms of cost, savings of \$2.03 per month was recorded with the *Firewood Jambar* [23] cookstove (Tables 3 and 5).

**Time poverty (Cooking and fuel collection time savings).**   Reduction in cooking times was reported in 14 of the 20 studies that evaluated the time efficiency of ICS. The savings compared to the TCS/TSF ranged from 2min/day (RTI-TECA) [21] to 110 min/day (Ph*i*lipsHD4021LS) [25]. With fuel type, time savings was higher (22min/day) with firewood than with charcoal (5min/day) with the *Philips* ICS within the same study [20]. Although an increase of 17min/day was associated with increased loading, reloading, and lighting time of firewood with the *Gastov* [60], there was a 9min/day savings with the same ICS with crop residues [54]. Savings in fuel-gathering times were highlighted in all five self-reported accounts [2, 22, 23, 25, 56] with the highest reduction of 58.5% (414 min/week) and 46.5% (614 min/week) with *Kuniokoa* [49, 56] and *Save80* [22] *ICS*, respectively. Increase were reported of 2min/day, 5min/day, 17min/day, and 60 min/day with *Prakti-Leo* [21], *Hifadhi* [60], *Gastov* [60], and *RocketMud* [61] cookstoves, respectively.

**Supporting measures used in addition to the ICS Intervention.**   Twelve studies mentioned instituting additional measures alongside the ICS intervention, which varied in type and description. Of the eight studies that mentioned community awareness or educational approaches, only one study [5] described the educational content (dangers of exposure to smoke). While the study [47] reported encouraging and supporting participants with behavioural and environmental modifications, these targeted lighting and fuelling the ICS. Other additional measures included a support system (implementation team's contact details and posters with instructions) [63], follow-up visits aimed at troubleshooting the ICS [49, 54], repairs and replacement of cookstoves and repair parts [25, 59]), and community participation [56, 59, 60]. Despite the description of these additional measures, only one paper reported on the impact of the measure in relation to improved adoption of the *Gastov* ICS [54].

## Result section B: User perspective of cookstoves

This section describes the views of the end users of the improved cookstoves.

**Selection of cookstove brand and studies.**   Of the 23 cookstove brands reported in section A, only six met the requirements for our review questions 1 and 2, i.e. (availability, affordability, reduction of pollutants and/or improvement in health outcomes). In Fig 5, we describe our systematic selection of these cookstoves. Cookstoves were excluded if there was no reduction

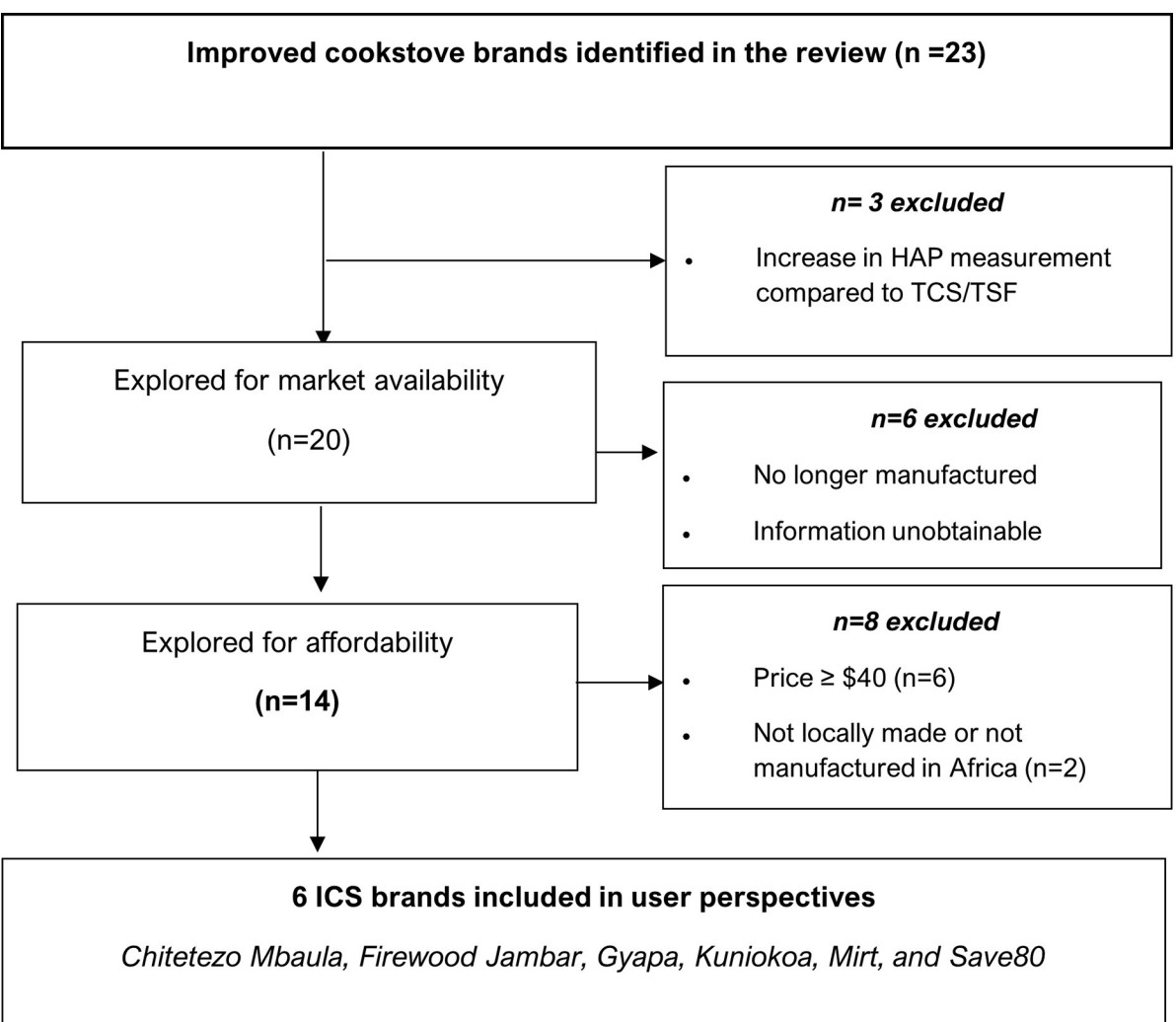

**Fig 5. Systematic selection of ICS for effectiveness in HAP reduction, market availability, and affordability.**

in HAP, no health benefit, not available, and not affordable. The ten articles selected to inform the user perspectives of the six cookstoves (*Chitetezo Mbaula* [64, 69, 70], Mirt [53, 68], *Firewood Jambar* [23], *Gyapa* [67], *Kuniokoa* [56], *Save80* [44, 45]*)* is presented in the PRISMA diagram (Fig 6).

**User perspectives.** Overall, users liked the ICS and found several advantages to their use compared to the TCS/TSF. For example, in all the studies, participants reported significant time-saving benefits with the ICS from cooking and fuel collection [64, 69], with food cooking faster [64, 67], and less time required to supervise the stove [23, 44, 45, 64]. This allowed participants to multi-task [23, 56] and resulted in more time for leisure and social activities [56].

Cooking pots were described as 'cleaner' with the *Gyapa* [67], offering timesaving from cleaning off black soot associated with TSF. Interestingly, and related to timesaving, users of the *Kuniokoa* cookstove highlighted a more equitable distribution of cooking duties with "male partners helping with cooking tasks" [56].

Some participants described 'family togetherness', that "the change has been positive to me and my family because they like the Kuniokoa and this has made us feel that we belong to a

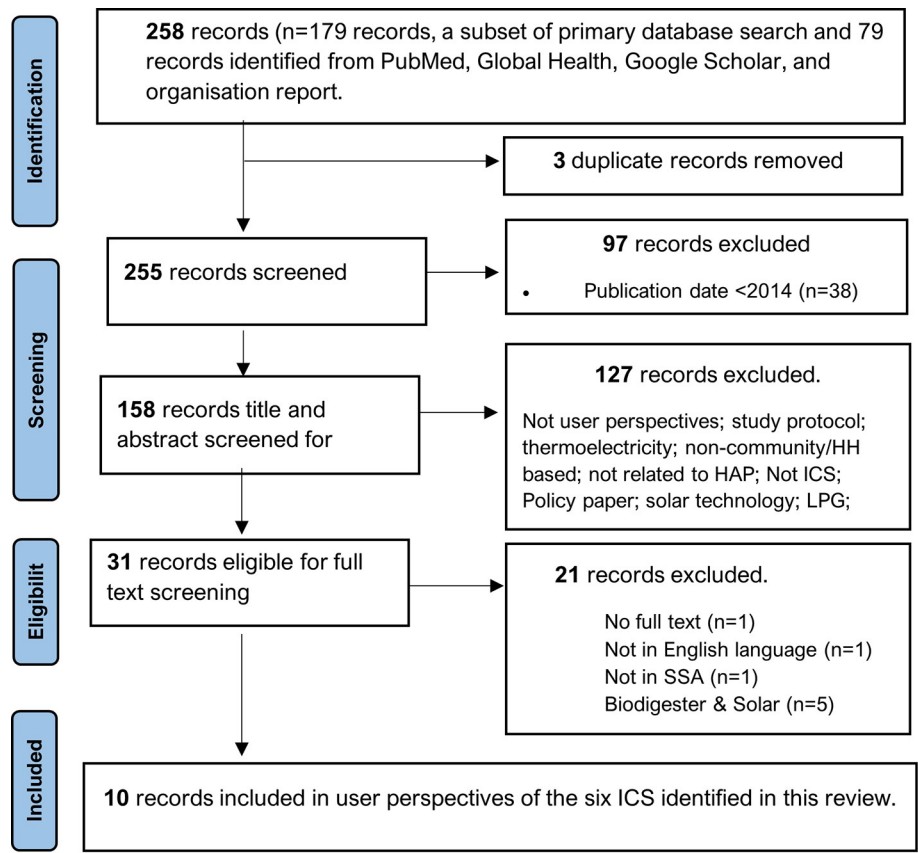

Abbreviation. HAP: Household Air Pollution, LPG: Liquid Petroleum Gas, SSA: Sub-Saharan Africa, ICS: Improved Cookstoves

**Fig 6. PRISMA flow diagram showing the selection of sources for User perspectives.**

developed family" [56], suggesting a subtle link between wellbeing and ICS use. Users also reported reduced back pain [70] from less bending and fuel collection [56] and less eye irritation [67]. Other reported advantages of the ICS included reduced fuel use [69, 70], suitability of use for various cooking tasks [69, 70], stoves being lightweight, portable and durable [67, 69], affordability [69, 70], and better shielding from wind and dust [23, 64] compared to baseline TCS/TSF.

Burns injury used as a proxy for safety in our review was viewed as a highly significant advantage of the improved cookstoves in the 'willingness to adopt' study [69], where over 95% (n = 121) of participants reported reduced incidence of burns and accidents. However, while the covering chamber for the flame was reported as a good design and safety measure [70], users also commented on excessive heat from the chamber and a lengthy cooling period, "making it unsafe for children" [70]. Other ICS disadvantages include reduced durability [67], unsuitability of the burner size for large cooking pots used in large-size households [23], resulting in stove stacking to meet the cooking needs, and back pain was associated with the low-height design of *Chitetezo Mbaula* [69]. In addition, compared to readily available firewood used with TSF, the fuel cost was higher with ICS [69], with igniting taking longer [70]. A thematic description of the user perspective is presented in Fig 7

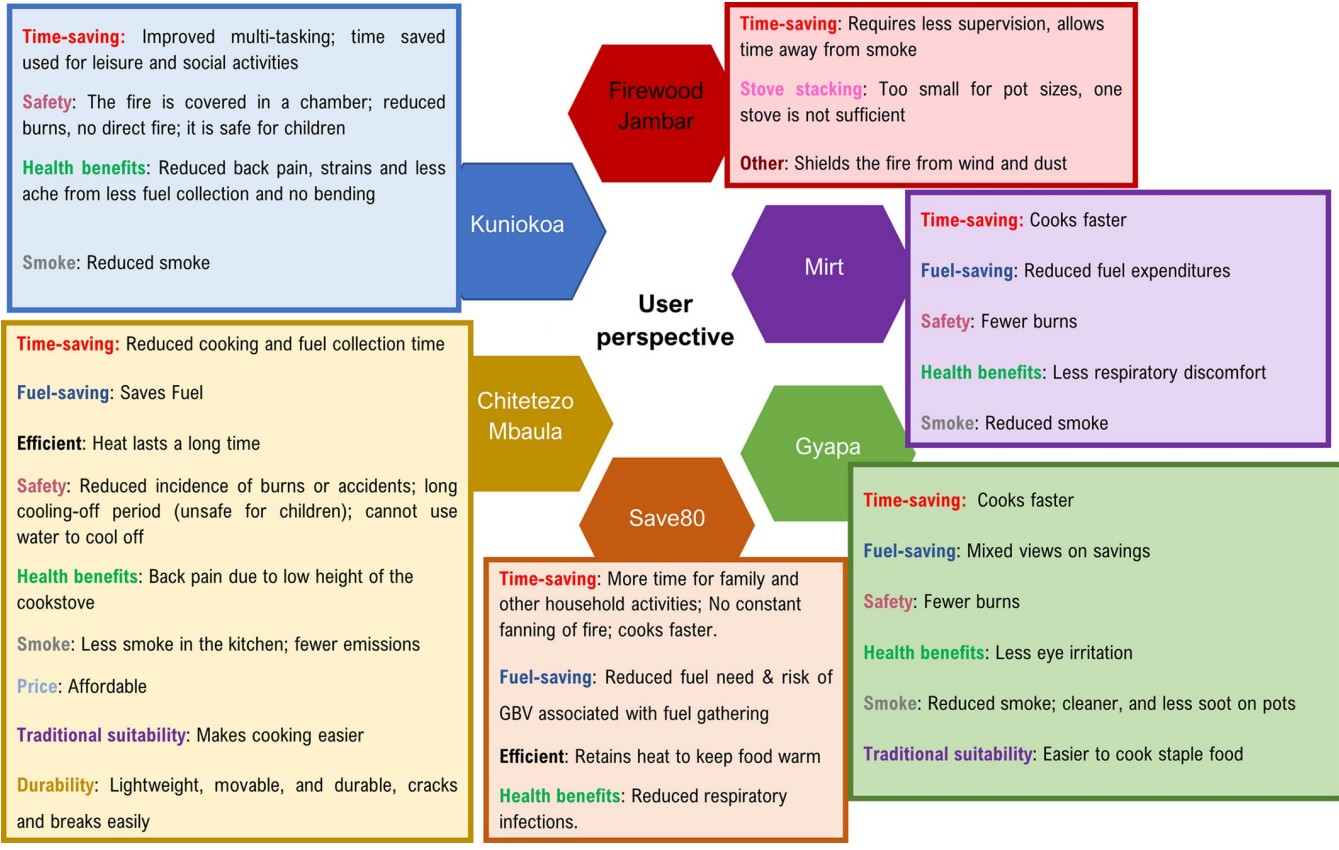

**Fig 7. Thematic presentation of user perspectives of available, affordable, and efficient improved cookstove.**

### Benchmarking the International Workshop Agreement (IWA) tier of cookstove performance with available field data

The International Organization for Standardization (ISO) IWA uses four indicators (efficiency, total emissions, indoor emissions, and safety) to evaluate cookstove performance across five tiers (0–4), with Tier 4 as the highest and Tier 0 as the lowest-performing cookstoves [36]. These indicators are based on performance under laboratory conditions. Table 6 presents an overview of the current IWA tier rating alongside the collated field data from this review. The field values examined in this review are subject to factors such as the cooking area's ventilation level and the fuel's water content which generally were not described within the studies reviewed. Therefore, we recommend caution in the use and interpretation of the scorecard.

## Discussion

This systematic scoping review identified and examined 39 field-tested ICS from 31 intervention studies and two non-peered review reports published between 2014 to 2022 in sSA. The aim was to identify available, affordable, safe, and efficient ICS, effective in reducing harmful emissions compared to TCS/TSF in real-life settings and able to meet user cooking needs.

Identifying only 33 intervention studies across nine of the 42 sSA countries reflects the paucity of available field evidence. It highlights the need for more field-testing of ICS in sSA, the region with the highest reliance on biomass and, therefore, exposure to household pollutants [10, 72]. The review also demonstrates the wide heterogeneity in the descriptions of ICS

**Table 6. Available cookstoves by IWA tier and field emissions score card.**

| Cookstove | Cost (USD) | Stove description | IWA Tier | | | | Field-based results from this review (% reduction) | | | Field-base health outcome from this review |
|---|---|---|---|---|---|---|---|---|---|---|
| | | | Indoor emission | Efficiency | Safety | Sustainability | Kitchen | Personal | | |
| | | | | | | | $PM_{2.5}$ | $PM_{2.5}$% | CO % | |
| **Tier four** | | | | | | | | | | |
| Mimi moto | 40 to 65 | Portable, Gasifier (TLUD), Fan, Solar Panel, pellets, wood | 4 | 4 | - | 10 years | - | -97 | - | Reduced systolic and diastolic BP Reduced shortness of breath Reduced burns |
| **Tier three** | | | | | | | | | | |
| Ace 1 | 90 | Gasifier, Fan (TLUD), dung, pellets, crop residue, portable | 3 | 3 | 4 | 12 years | - | -40 | - | - |
| Kuniokoa | 30 to 41 | Rocket, Portable, Side-feed | 3 | 2 | 3 | 3 years | - | - | - | Decrease smoke inhalation. Decrease intense heat. Less back strain from bending to blow on fire. At least 8hrs/wk. increase in sleep time Reduced burns |
| **Tier Two** | | | | | | | | | | |
| Biolite | 40 to 70 | Fan, TEG, Portable, crop residue, dung, wood | 2 | 2 | - | 5 years | - | - | 1.5 | - |
| **Tier One** | | | | | | | | | | |
| Envirofit | 100 | Rocket, Portable, wood fuel | 1 | 2 | 3 | - | -36 | -27 | -54 | - |
| **Tier Zero** | | | | | | | | | | |
| Gyapa woodstove | 7 | Rocket, Batch load, Portable, wood | 0 | 2 | - | 3 years | -18 | -21 | - | Reduced pneumonia cases |

(*Continued*)

**Table 6.** (Continued)

| Cookstove | Cost (USD) | Stove description | IWA Tier | | | | Field-based results from this review (% reduction) | | | Field-base health outcome from this review |
|---|---|---|---|---|---|---|---|---|---|---|
| | | | Indoor emission | Efficiency | Safety | Sustain ability | Kitchen | Personal | | |
| | | | | | | | PM$_{2.5}$ | PM$_{2.5}$% | CO % | |
| **Unrated** | | | | | | | | | | |
| Chitetezo-Mbaula | 2–4 | Rocket, Side-feed, wood Crop residue, Portable, | - | 2 | 3 | 4 years | -13 | -8.2 | -33 | Reduced cough, SOB Reduced backpain Reduced eye burning Increase sneezing and running nose Increase cases of burns |
| EcoZoom Dura | 30–40 | Rocket, side-feed, chimney, | - | - | - | 5 years | -46 | - | - | Reduced burns |
| EcoZoom | 30–40 | Rocket, side-feed, no chimney | - | - | - | - | -20 | 2 | -32 | - |
| Mirt Stove | 3.5 to 7 | Rocket, portable, chimney, dung, briquettes, wood | - | - | - | 5 years | -10 | - | | Reduced smoke Less respiratory discomfort Reduced respiratory illness in children under age 5 years. |
| Save80 | 20 to 55 | Portable, Pot skirt, Sunken pot, Crop residue, Dung, wood | - | - | - | 15 years | - | - | -96.4 | Reduced cough Reduced sore eyes |
| Eco Chula | 29 to 33 | Gasifier (TLUD), Portable, Wood | - | - | - | - | -18 | -21.5 | -68 | - |
| Firewood Jambar | 10 | Ceramic-lined, wood | - | - | - | 2 years | - | - | - | Reduced eye problems Reduced respiratory problems |

(*Continued*)

**Table 6.** (Continued)

| Cookstove | Cost (USD) | Stove description | IWA Tier | | | | Field-based results from this review (% reduction) | | | Field-base health outcome from this review |
|---|---|---|---|---|---|---|---|---|---|---|
| | | | Indoor emission | Efficiency | Safety | Sustainability | Kitchen | Personal | | |
| | | | | | | | PM$_{2.5}$ | PM$_{2.5}$% | CO % | |
| Hifadhi | - | Crop residue, wood | - | - | - | 5 years | - | 11.1 | - | - |
| Prakti Leo | 23.62 | Chimney, Multiple burners, Rocket, wood | - | - | - | 5 years | -39 | -32.3 | -45 | - |

characteristics, designs, supporting interventions, and metric units (PPM, mg/m$^3$, μg/m$^3$, g/kg) used in intervention studies. The heterogeneity prevented effective comparison of HAP emission levels across the different ICS and was further compounded by the scant reporting of HAP-associated factors such as fuel water content for units measured in g/Kg, cookstove features (unspecified chimney and draft systems), and household structure (e.g., ventilation, chimney, open or enclosed roof) in the studies.

## Emission levels, health, and safety-related outcomes

Our main finding shows a general trend of ICS in reducing PM$_{2.5}$ compared to baseline TCS/TSF. However, no ICS was effective in reducing levels close to the WHO-IAQ safe level, with the lowest reduction of 0.11mg/m$^3$ over four times the WHO 24-hour average of 0.025mg/m$^3$ for safe indoor air quality. In contrast, CO levels were substantially reduced (lowest value of 0.2PPM) below the WHO recommendations of 6.11PPM. These findings are reported in previous reviews [7, 27, 28], which also found that ICS interventions did not reduce PM$_{2.5}$ close to the WHO interim guideline levels.

In line with emission reduction, the forced-draft cookstoves consistently show the highest reduction levels of personal and household PM$_{2.5}$ and CO. This accords with Memon et al.'s systematic review [73] findings on the effectiveness of forced-draft cookstoves in reducing incomplete fuel combustion, thereby reducing exposure to harmful emissions [36, 74]. Kumar et al.'s [7] meta-analysis also reported the lowest reduction in kitchen-level CO with advanced combustion cookstoves. Regarding ICS chimney features, our findings of a higher reduction in kitchen PM$_{2.5}$ levels with the chimney cookstove *EcoZoom* Dura [59] compared to the same brand non-chimney *EcoZoom* ICS [21] concurs with earlier systematic reviews evidence [7, 27] that chimneys play a role in reducing HAP emission. However, when compared across ICS brands, we found this association inconclusive, with some ICS brands without chimneys reporting higher emission reductions than chimney cookstoves. A possible explanation for this might be related to the chimney's primary function of directing smoke away from the cooking areas, usually into the ambient environment [75] and the chimney heights, which were not described in any of the included studies. Some chimneys, for example, direct smoke away from the cookstove but do not remove it from the kitchen, which could result in a higher concentration of pollutants, especially in kitchen areas with minimal or no ventilation.

Further, the practice of measuring personal CO and personal $PM_{2.5}$ levels compared to household-level measurements (function of cookstoves and fuel used) will produce differing results. Dickinson et al. [76] describe how the 'function of activity' measured by personal exposure does not capture accurate data such as duration of cooking and type of dishes prepared. These activities are dependent to some extent on user behaviour, such as the actual time spent in the kitchen or how closely the cooks stand to the cookstove while cooking and may not reflect the function of the cookstove. This can also be deduced from the users' account of spending less time supervising the fire, suggesting reduced exposure time to pollutants. While personal level measurement may not capture accurate data on ICS effectiveness, it could inform data on the users' behaviour of spending less time in the fire's proximity if examined together with kitchen-level emissions in the same setting.

In addition to emission levels, health-related outcomes provide another measure of the effectiveness of the ICS in reducing health risks and improving user experience. Measured (BP), self-reported eye and respiratory symptoms, and user accounts of back pain, burns, and smoke, reported a reduction in health-related symptoms irrespective of levels of $PM_{2.5}$, CO, and black carbon in the reviewed studies. Previous systematic reviews have demonstrated reductions in respiratory and non-respiratory health [28] and burn injuries [18]. There are also similarities between the effect of ICS on blood pressure expressed in studies reviewed and those described by Onakomaiya et al. [26] and in Kumar et al.'s metanalysis review [7] of statistically significant BP reductions despite pollution reduction not being at safe levels.

## Affordability and availability

The use of less harmful cookstoves can only be realised in practice if the cookstoves are available and affordable to the end user. Some studies alluded to high costs contributing to low ICS adoption rates, especially in more disadvantaged communities [16, 18, 77]. Notably, within this review, only a few ICS with reduced emission levels were priced at less than $40, a cost subject to inflation with additional shipping and importation fees [37, 78]. While we adopted a <$40 price cap in this review due to the focus on the poor communities in sSA, we acknowledge that even a price of $40 would require subsidisation, with almost 40% of 1.08 billion people in sSA living below 1.90 per day [79]. Even with the subsidisation, the purchasing power parity, e.g., $1≈312.3 of Malawi Kwacha [80] of most sSA currencies, could compound the affordability issue, resulting in very limited affordable ICS options for the poorer households and creating a practical barrier to ICS accessibility and adoption [15]. However, the study, which subsidised the ICS price to enhance adoption, found that the approach failed to address the long-term affordability of ICS for poor households [22]. Similarly, in Rosenbaum and colleagues' 'willingness to pay' study [17], of the 105 participants, only one opted to buy the ICS at market price ($19-$54) and most reported a preference to keep the ICS if it were free or available at a nominal price. Additional to the price of the ICS, reductions in the fuel used (where purchased) and in timesaving (where gathered) could be translated into an economic benefit. This is mentioned in several studies [15, 22, 65, 81], where fuel reduction, if measured in time saved, would result in enhanced productivity to invest in other income-generating activities. It also accords to the report from systematic reviews [15, 70], where participants ranked the importance of fuel savings higher than smoke emissions reduction.

Although we did not examine fuel unit prices in this review, their significance is not to be underestimated in determining the feasibility and effectiveness of ICS in poor communities. Poor households, for instance, are unable to obtain wood pellets due to high prices and limited availability, although there is evidence that they produce more heat, reduce emissions, and

improve health outcomes [49]. Therefore, wood pellet availability and cost reduction would be required to increase ICS options for most poor rural communities.

## Cookstove durability

Only four of the 33 studies in this review described the cookstoves' durability/ sustainability. A high incidence of cookstove breakdowns will discourage communities from using ICS. More-over, for households unable to afford a replacement, this could facilitate the reverting to inefficient cookstoves to meet their cooking needs. Therefore, field studies exploring the uptake and sustainability of ICS should also consider stove reliability in addition to its cost and ease of stove repair where required.

## Gender equality

Few studies in this review linked findings of time spent on fuel collection and cooking with gender, with only one study suggesting a direct association between ICS use and observed improvement in the health of women who are primary cooks [23]. This disproportionate gender impact of HAP from inefficient cookstoves is referred to as 'an obstacle to women's human rights, health and sustainable development by Hyde and colleagues [82] and requires urgent and greater attention. Furthermore, the continued promotion and use of inefficient cookstoves further perpetuates the cycle of gender-related ill-health and poverty associated with HAP [83] from long and unpaid hours (≈14 hours/day) of women and girls in developing countries [2] undertaking household chores such as fuel gathering and meal preparation [2, 14].

## User perspectives

Finally, our exploration of user perspectives highlights the importance of user experience in promoting the scale-up of cleaner cookstoves. Users suggest that they value less time required for supervision, less time cleaning soot off pots, multitasking opportunities, and more time for families to socialise. User accounts focus on the health-related benefits of ICS more widely than specific symptoms reported in many studies to improve feelings of well-being. Also, while most cookstove designs are centred around reducing emission levels and fuel use, users suggest that other factors, such as ICS height for postural comfort and the convenience of moving the cookstoves to different locations, are important. The ICS portability would particularly be valued in households with a shared cooking and living space due to higher accumulation and increased exposure to pollutants in shared spaces [11]. In addition, cultural cooking practices also play a significant part in stove satisfaction, with 'suitability for cooking a traditional meal' seen by users as an important benefit of an ICS. These practices are also reflected in the preference for 'burners to fit large or multiple cooking pots' for large family sizes and/or communal cooking in most local communities in sSA. The users also called attention to the safety of four of the six ICS explored for user perspectives. In addition to fewer burns, the combustion chamber's importance in enclosing the flame made it safer for children. However, a more prolonged cooling-off period and inability to cool with water were reported as unsafe by some users of the *Chitetezo Mbaula*.

In summary, good stove design, fuel and time savings, health benefits, and meeting traditional cooking needs have been identified as critical to cookstove uptake [15]. It is essential in this review to highlight that most of the themes described by the users centred around the cookstoves' design, indicating the value of including the end-user voice at the design stage, which could address some of the root causes of stove-stacking identified in several other studies [15, 16, 43]. For example, a brick plinth against the wall could raise the height of Mbaula,

direct the smoke nearer to the ceiling vents (ventilation), reduce back pain, and raise it above child level, thereby reducing the risk of burns.

## Strengths and limitations

Collating relevant recent evidence on all ICS that have reduced harmful emissions in the field, alongside information on availability and affordability, whilst also considering user cooking needs, is an important step forward in ICS assessment. To our knowledge, this is the first field evidence collated together in this way. Our approach to the review's questions, aims, search strategy and reporting was systematic and guided by an established scoping review framework [34]. The validity of the search outcome was also enhanced by hand searches of reference lists and studies reported within identified systematic reviews. Our quality appraisal process (not mandatory in scoping reviews) allowed evaluation of the quality of evidence available for ICS field studies. All reviewed studies had pre-intervention exposure to TSF or traditional cookstoves for comparison across studies. In addition, we reported on measures such as awareness and support alongside ICS promotion, albeit limited studies described these measures. Finally, the additional exploration of the user voice gives a deeper understanding of ICS features that are important to the communities most likely to benefit from these stoves. While it reflects the importance of the voice of the end-user, it highlights the gap in evidence available based on the user voice in HAP interventions.

## Conclusion

While ICS have increased in popularity in recent decades as an alternative to the three-stone fire or traditional stove, their characteristics and effectiveness in reducing HAP differ considerably. However, given their importance as an interim solution until global access to clean energy sources and cleaner cooking technologies is achieved, the scale-up of ICS needs to be underpinned by evidence of substantive reductions in HAP compared to many currently being used. Based on the findings of this review, the following recommendations are suggested to inform research, policy, and current practices in the design and promotion of ICS in SSA (Table 7)

Table 7. Recommendation for research, practice, and policy.

| Recommendations | Target |
|---|---|
| • Given that evidence suggests that ICS reduce HAP but not close to safe levels, additional interventions should be promoted alongside cleaner stoves as standard practice. Future research should identify any additional benefits of community engagement practices, cleaner lighting sources, adequate storage for drying of wood, and improved ventilation alongside ICS. | Research and Policy |
| • A detailed standardised description of all relevant study information should be considered the gold standard for the field cookstove evaluation study. This should include cookstove design, chimney height, variation in kitchen design (type of ventilation), availability and cost to the local user, sustainability and durability, ease of repair, incidence of burns, user perspectives, season, i.e., wet or dry, and detailed description of type fuel including water content. | Research and practice |
| • The use of available local resources, such as knowledge, skills, and raw materials, should be considered when developing an ICS to reduce purchasing costs and enhance the community's skills and capacity to maintain the ICS with less reliance on the importation of repair parts, thereby increasing the lifespan of the intervention. | Research and practice |
| • ICS engineers and researchers should ensure that the user perspective informs all stages of ICS development. The user voice will ensure that ICS meets household socio-economic, cultural, gender, and structural needs. | Policy, Research and Practice |
| • Policy and funding bodies should place more emphasis on the assessment of cookstove efficiency and net health benefits before promoting them to poor communities. | Policy |

## Supporting information

**S1 Table. Preferred Reporting Items for Systematic reviews and Meta-Analyses extension for Scoping Reviews (PRISMA-ScR) checklist.**
(DOCX)

**S2 Table. Available descriptions of ICS examined in this review.**
(DOCX)

**S1 Fig. A.** Sample of database search terms with results from EMBASE June 2020, July 2021, September 2022**. B.** Sample of relevant organisation searches and outcomes.
(ZIP)

**S2 Fig. Study's data extraction tool- Excel®.**
(TIF)

**S3 Fig. Data extraction tool for users' perspective.**
(TIF)

**S4 Fig. A.** Sample of quality appraisal of included quantitative study using LQAT and the global rating tool. **B.** Sample of quality appraisal of included qualitative study using adapted Hayden et al.'s and the global rating tools.
(ZIP)

## Acknowledgments

We acknowledge and appreciate the support, time, and expertise of Paul Murphy, the information specialist at the Royal College of Surgeons in Ireland library, with the literature searches. pjmurphy@rcsi.ie, ORCID 0000-0001-5056-1971.

## Author Contributions

**Conceptualization:** Eunice Phillip, Aisling Walsh, Mike Clifford, Debbi Stanistreet.

**Data curation:** Eunice Phillip, Jessica Langevin, Megan Davis, Nitya Kumar, Aisling Walsh, Vincent Jumbe, Debbi Stanistreet.

**Formal analysis:** Eunice Phillip.

**Investigation:** Eunice Phillip, Jessica Langevin, Megan Davis, Nitya Kumar, Debbi Stanistreet.

**Methodology:** Eunice Phillip, Aisling Walsh, Debbi Stanistreet.

**Project administration:** Eunice Phillip.

**Resources:** Ronan Conroy.

**Supervision:** Eunice Phillip, Debbi Stanistreet.

**Writing – original draft:** Eunice Phillip.

**Writing – review & editing:** Eunice Phillip, Jessica Langevin, Megan Davis, Nitya Kumar, Aisling Walsh, Vincent Jumbe, Mike Clifford, Ronan Conroy, Debbi Stanistreet.

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
