## [Decision Letter · Decision Letter 0]

6 Mar 2023

PONE-D-22-35350Improved cookstoves to reduce household air pollution exposure in sub-Saharan Africa: a scoping review of intervention studies.PLOS ONE

Dear Dr. Phillip,

Thank you for submitting your manuscript to PLOS ONE. After careful consideration, we feel that it has merit but does not fully meet PLOS ONE’s publication criteria as it currently stands.  Reviewer of your manuscript has suggested minor revision on your submission. Therefore, we invite you to submit a revised version of the manuscript that addresses the points raised during the review process. 

We look forward to receiving your revised manuscript.

Kind regards,

Srijan Lal Shrestha, Ph.D.

Academic Editor

PLOS ONE

Journal Requirements:

2. We note you have included a table to which you do not refer in the text of your manuscript. Please ensure that you refer to Table 7 in your text; if accepted, production will need this reference to link the reader to the Table.

3. We note that this manuscript is a systematic review or meta-analysis; our author guidelines therefore require that you use PRISMA guidance to help improve reporting quality of this type of study. Please upload copies of the completed PRISMA checklist as Supporting Information with a file name “PRISMA checklist”.

Reviewers' comments:

Reviewer's Responses to Questions

**Comments to the Author**

1. Is the manuscript technically sound, and do the data support the conclusions?

Reviewer #1: Yes

2. Has the statistical analysis been performed appropriately and rigorously? 

Reviewer #1: N/A

3. Have the authors made all data underlying the findings in their manuscript fully available?

Reviewer #1: Yes

4. Is the manuscript presented in an intelligible fashion and written in standard English?

Reviewer #1: Yes

5. Review Comments to the Author

Reviewer #1: The paper based on systematic review is well written, and is attempted to identify which improved cookstoves would be the most suitable to promote among poor communities in Sub-Sahara Africa (sSA).The manuscript is technically sound, systematic review is done following the standard protocol, quality assessment of the included studies are also assessed by using standard tools. The conclusions are supported by the data. Data curation is done rigorously. However there are still some comments to improve the manuscript, which are as follows.

1. In Line 171 under 'Data Charting', it is mentioned that only the most comprehensive study data were included where multiple publications reported on the same data. What is the basis used for considering them the most comprehensive? It is suggested to explain point wise clearly.

2. In Line 225, it is reported that the total number of sSA countries is 46. Again in contrary with this, the total sSA countries is reported 42(in Line 507). Actually the total number of sSA countries is more than 46. Please make your reporting consistent, and report exact number of sSA countries what exactly is in existence.

3. Add percentage along with numbers in Figure 1 related to sparsity of reported relevant studies in Sub-Saharan Africa. It will help readers to understand quickly.

4. If possible, present the major review report according to some geographic distribution of Sub-Saharan Countries such as West Africa, East Africa, Central Africa, and Southern Africa which suits for the analysis based on the studies incorporated. There may be considerable variations with respect to population distribution, socio-economic indicators and others. The analysis based on some appropriate geographical domain may be helpful for future policy point of view.

5. Figure numbers are not matching in text and in the provided figures, and they are not in sequential order. Some figures indicated in the text such as (S2 & S3 Figs) not found in the provided figure lists. It is strongly suggested to check all figure numbers indicated in the text and keep accordingly.

6. In each table, please add serial number creating a first column (if possible) so that it would be easy for the readers to go through it.

7. In Table 2, only 'Section B' is mentioned without mentioning 'Section A'. Please add Section A in appropriate place of the table.

8. Check all the symbols used in Table3, and match with Reference Key. There is mismatch in the notations such as 'NM= Not measured', 'NM=Not measured in the study'. What is the difference between them? In Table 5 also, A study namely LaFave 2021[57] RCT- 10th column, what is the meaning of "?". If it is symbol, please change by another one. I suggest you to go through them one by one.

9. Please add statistical method/statistical test/statistical model whichever was applied in each study incorporated in this review report in Table 3 and Table 5 where estimates and p-values are reported. In the manuscript, it is only reported different estimates, p-values, and confidence interval, etc. which are not comprehensive.

10. Figure 7 seems to be prepared through some step function using time to event data (having censored data) because in some costing points, there is constant probability and then increases. Generally such type of curves can be observed in hazard function. It is better to explain the curve for making it comprehensive.

11. Add the definition of quality assessment criteria ('strong', 'moderate', & 'weak)' in the text for clarity though it is indicated in the submitted supplementary document.

6. PLOS authors have the option to publish the peer review history of their article (what does this mean?). If published, this will include your full peer review and any attached files.

Reviewer #1: No

---

## [Author Response · Author response to Decision Letter 0]

7 Apr 2023

Editor's comment 1. Please ensure that your manuscript meets PLOS ONE's style requirements, including those for file naming.

Authors: Thank you. We have followed the provided guidelines throughout the manuscript, including formatting styles, authors’ names, addresses and tags. 

Editor's comment 2. We note you have included a table to which you do not refer in the text of your manuscript. Please ensure that you refer to Table 7 in your text; if accepted, production will need this reference to link the reader to the Table.

Authors: Thank you for pointing this out. We have referenced the table in the manuscript text. Page 46.

Editor's comment 3. We note that this manuscript is a systematic review or meta-analysis; our author guidelines, therefore, require that you use PRISMA guidance to help improve reporting quality of this type of study. Please upload copies of the completed PRISMA checklist as Supporting Information with the file name “PRISMA checklist”.

Authors: We submitted Preferred Reporting Items for Systematic Reviews and Meta-Analyses Extension for Scoping Reviews (PRISMA-ScR) Checklist as part of the supplementary information with the prior submission. We have updated the form to correspond to the changes made in the manuscript. S1 Table…. We submitted Preferred Reporting Items for Systematic Reviews and Meta-Analyses Extension for Scoping Reviews (PRISMA-ScR) Checklist as part of the supplementary information with the prior submission (S1 Table). We have updated the supplementary information file to correspond to the changes made in the manuscript. We’ve submitted tracked and clean versions of the SI file. 

Editor's comment 4 Please review your reference list to ensure that it is complete and correct. If you have cited papers that have been retracted, please include the rationale for doing so in the manuscript text or remove these references and replace them with relevant current references. Any changes to the reference list should be mentioned in the rebuttal letter that accompanies your revised manuscript. If you need to cite a retracted article, indicate the article’s retracted status in the References list and include a citation and full reference for the retraction notice.

Authors: We have reviewed and formatted the reference list to match the journal’s criteria. We changed the reference (United Nations Development Program. About Sub-Saharan Africa: United Nations; 2021) to The African Union Commission—member states to accurately address reviewer’s comment 2 below. 

Reviewer’s comments

1. In Line 171 under 'Data Charting', it is mentioned that only the most comprehensive study data were included where multiple publications reported on the same data. What is the basis used for considering them the most comprehensive? It is suggested to explain point-wise clearly.

Authors: Thank you for the comment. We compared (1) the population studied to see if they were the same. (2) If not, we included both studies. If yes, we looked at what was measured, analysed, and reported. In the one study that we excluded, the same population was sampled for kitchen PM2.5 & CO, and a mother and one child’s personal CO over 48 hours using six cookstove interventions. This was the same as the authors' previous publication except that the study included in our review analysed and reported the same result but included confounders in the analysis such as fuel used for lighting, number of cooking episodes, and average number of people cooked for. We have rephrased and included a succinct description of this step in the manuscript (Page 7)

2. In Line 225, it is reported that the total number of sSA countries is 46. Again in contrary with this, the total sSA countries is reported 42 (in Line 507). Actually the total number of sSA countries is more than 46. Please make your reporting consistent, and report exact number of sSA countries what exactly is in existence.

Authors: We thank you for your observation. We have updated this with the number of SSA countries (48) reported by the African Union and updated the reference source to reflect this. 

3. Add percentage along with numbers in Figure 1 related to sparsity of reported relevant studies in Sub-Sahara Africa. It will help readers to understand quickly.

Authors: Thank you for this comment. We have changed Figure 1 to show the high number of countries without relevant studies and the percentage of studies in the countries with relevant studies. 

4. If possible, present the major review report according to some geographic distribution of Sub-Saharan Countries such as West Africa, East Africa, Central Africa, and Southern Africa which suits for the analysis based on the studies incorporated. There may be considerable variations with respect to population distribution, socio-economic indicators and others. The analysis based on some appropriate geographical domain may be helpful for future policy point of view.

Authors: Thank you for your comment. We agree that collating the evidence based on the geographical distribution would be relevant to policy and enhance the manuscript. However, the sparsity of description of information like socioeconomic factors, food cooked and exposure levels in almost all the studies makes synthesising the data using the SSA geographical distribution. In addition, there are too few studies in each SSA geographical distribution, and the diverse population studied would not allow comparison. This type of analysis would be best suited for systematic review, focusing on specific parameters across the 4 regions in SSA.

5. Figure numbers are not matching in text and in the provided figures, and they are not in sequential order. Some figures indicated in the text such as (S2 & S3 Figs) not found in the provided figure lists. It is strongly suggested to check all figure numbers indicated in the text and keep accordingly.

Authors: Thank you for your comment. The indicated S2 and S3 (data extraction tools) you highlighted were included in our previous supplementary information (SI)file submission. We have ensured that the listed SI list at the end of the manuscript matches the named tables and figures. We have collated all the SI figs and tables sequentially. 

Authors: Thank you for your comment. The indicated S2 and S3 (data extraction tools) you highlighted were included in our previous supplementary information (SI)file submission. We have ensured that the listed SI list at the end of the manuscript matches the named tables and figures. We have collated all the SI figs and tables sequentially. 

6. In each table, please add serial number creating a first column (if possible) so that it would be easy for the readers to go through it.

Authors: We see the rationale in your comment. However, it was only possible to add an extra column to Table 2. We could not achieve this with Table 3 without it being out of margin. We have added the associated reference to the ICS names to make it easier for readers.

7. In Table 2, only 'Section B' is mentioned without mentioning 'Section A'. Please add Section A in appropriate place of the table.

Authors: Thank you. We have included section A and its label in Table 2. (page 11)

8. Check all the symbols used in Table3, and match with Reference Key. There is mismatch in the notations such as 'NM= Not measured', 'NM=Not measured in the study'. What is the difference between them? In Table 5 also, A study namely LaFave 2021[57] RCT- 10th column, what is the meaning of "?". If it is symbol, please change by another one. I suggest you to go through them one by one.

Authors: Many thanks for this observation. We have corrected and updated the key used in all the tables to reflect their intended meanings. NM now corresponds to ‘Not measured in the referenced study’. The "?" in Table 5, 10th column was an error. This has now been corrected, and the sentence, adjusted for clarification. All the symbols have been updated. 

9. Please add statistical method/statistical test/statistical model whichever was applied in each study incorporated in this review report in Table 3 and Table 5 where estimates and p-values are reported. In the manuscript, it is only reported different estimates, p-values, and confidence interval, etc. which are not comprehensive.

Authors: Thank you for your suggestion. We have included this information in Tables 3 and 5 under their respective sections. We denoted this with ^ = Statistical test (study #) in the ‘KEY’ sessions. We did not report in the tables as suggested due to table size challenges/limitations.

10. Figure 7 seems to be prepared through some step function using time to event data (having censored data) because in some costing points, there is constant probability and then increases. Generally, such type of curves can be observed in hazard function. It is better to explain the curve for making it comprehensive.

Authors: Thank you for your comment. We have assumed that the figure in question is Fig 4 (per cent of available cookstove price). The figure is simply the cumulative distribution function. To avoid readers puzzling over this term, we have captioned it "Stoves costing this much or less". The reason why the graph looks like a hazard function is that this, too, is a cumulative distribution but, of event probabilities, not stove prices. The graph is stepped rather than continuous as the price rises because the number of stoves only rises when a further stove becomes available for that amount of money. The revised submission has minor improvements in the manuscript and captioning, as noted above.

11. Add the definition of quality assessment criteria ('strong', 'moderate', & 'weak)' in the text for clarity though it is indicated in the submitted supplementary document.

Authors: Thank you for your comment. We have included the description of our rating under the method section under quality appraisal for clarity.

---

## [Editor Report · Decision Letter 1]

12 Apr 2023

Improved cookstoves to reduce household air pollution exposure in sub-Saharan Africa: A scoping review of intervention studies.

PONE-D-22-35350R1

Dear Dr. Phillip

We’re pleased to inform you that your manuscript has been judged scientifically suitable for publication and will be formally accepted for publication once it meets all outstanding technical requirements.

Kind regards,

Srijan Lal Shrestha, Ph.D.

Academic Editor

PLOS ONE
---

## [Editor Report · Acceptance letter]

17 Apr 2023

PONE-D-22-35350R1 

Improved cookstoves to reduce household air pollution exposure in sub-Saharan Africa: A scoping review of intervention studies. 

Dear Dr. Phillip:

I'm pleased to inform you that your manuscript has been deemed suitable for publication in PLOS ONE. Congratulations! Your manuscript is now with our production department. 

Kind regards, 

on behalf of

Dr. Srijan Lal Shrestha 

Academic Editor

PLOS ONE